# MindSet: Vision. A toolbox for testing DNNs on key psychological experiments

## Abstract

Multiple benchmarks have been developed to assess the alignment between deep neural networks (DNNs) and human vision. In almost all cases these benchmarks are observational in the sense they are composed of behavioural and brain responses to naturalistic images that have not been manipulated to test hypotheses regarding how DNNs or humans perceive and identify objects. Here we introduce the toolbox *MindSet: Vision*, consisting of a collection of image datasets and related scripts designed to test DNNs on 30 psychological findings. In all experimental conditions, the stimuli are systematically manipulated to test specific hypotheses regarding human visual perception and object recognition. In addition to providing pre-generated datasets of images, we provide code to regenerate these datasets, offering many configurable parameters which greatly extend the dataset versatility for different research contexts, and code to facilitate the testing of DNNs on these image datasets using three different methods (similarity judgments, out-of-distribution classification, and decoder method), accessible via GitHub. We test ResNet-152 on each of these methods as an example of how the toolbox can be used.

## 1 Introduction

Deep neural networks (DNNs) provide the best solution for visual identification of objects short of biological vision, and many researchers claim that DNNs are the best current models of human vision and object recognition (Kubilius et al., 2019; Mehrer et al., 2021; Zhuang et al., 2020; Dicarlo et al., 2023; Golan et al., 2023). Key evidence in support of this claim comes from the finding that DNNs perform the best on various behavioural and brain benchmarks. In the case of behavioural benchmarks, models are assessed on how well they account for human (or macaque) errors in classifying a large set of objects (Rajalingham et al., 2018; Tuli et al., 2021; Hebart et al., 2023), or how well they predict human similarity judgements (Peterson et al., 2017; Battleday et al., 2020). In the case of brain benchmarks, models are assessed with regard to how well they predict brain recordings (e.g., single-cell responses or fMRI data) in response to a set of objects (Schrimpf et al., 2018; 2020; Hebart et al., 2023; Allen et al., 2022). The general assumption is that the better a model does at predicting the data, the more similar the model is to biological vision. For instance, the Brain-Score benchmark is described as "a composite of multiple neural and behavioural benchmarks that score any [artificial neural network] on how similar it is to the brain's mechanisms for core object recognition" (Schrimpf et al., 2018)

A common feature of most benchmark studies is that they treat the to-be-predicted data as observational. That is, there is rarely an attempt to predict the impact of experimental manipulation designed to test specific hypotheses about how human or machine vision works. Rather, observers perform a single task over a set of images that satisfies some general criterion, such as objects presented in isolation (Kriegeskorte et al., 2008), in naturalistic contexts (Peterson et al., 2017; Battleday et al., 2020; Allen et al., 2022; Hebart et al., 2023), or on a range of arbitrary backgrounds (Cadieu et al., 2014; Schrimpf et al., 2018). This approach is problematic because it is possible to make good predictions on these datasets even when models identify objects in a qualitatively different way from monkeys or humans (Dujmović et al., 2024). For example, if the images contain multiple diagnostic cues for object

classification (e.g., shape and texture both predict object category), then good predictions might be driven by different features than those that drive human object recognition – that is, predictions might be driven by confounds. For example, a DNN that classifies objects by texture might still be able to predict brain activations in a visual system that classifies objects by shape.

The standard way to rule out confounds in order to determine causal relations (e.g. inferring that DNNs learn brain-like representations) is to carry out experiments designed to rule out confounds as the basis of making good predictions. In fact, there is a large literature in psychology describing experiments designed to test specific hypotheses about how human vision works, but surprisingly, this literature is often ignored when modellers compare DNNs to biological vision (for notable exceptions see the review paper Kanwisher et al. (2023)). Bowers et al. (2023b) reviewed a wide range of psychological phenomena that current DNNs either fail to capture or that have yet to be considered. Furthermore, when researchers do consider the psychological literature when making claims regarding DNN-human similarities, the models are rarely subject to the kind of "severe" tests that are required to make any strong conclusions, that is tests that are likely to challenge claims in case they are false. Instead, strong conclusions are often drawn based on superficial similarities (Bowers et al., 2023a).

There are at least four (related) reasons for this. First, many researchers in computer science and computational neuroscience may be unfamiliar with the rich set of experiments carried out in psychology that manipulate independent variables to better understand human vision, memory, language, etc. Those who are aware of these studies might find it challenging to engage with them, as psychological datasets are not readily available in formats that the community is accustomed to working with. Second, it is not always obvious how one should test a model against psychological data. Hence, it may be easier to focus on improving performance on the current benchmarks, and this may have discouraged researchers from exploring data from psychology. A third potential reason is an overall skepticism towards psychological results, a sentiment that may reflect the well-documented replication crisis in psychology (Baker, 2015). Forth, there is a strong bias to look for DNN-human similarities and downplay the differences (Bowers et al., 2023a), and severely testing on psychological data might not result in similarities. However, characterizing these failures provides key insights into the ways DNNs need to be improved when modelling biological vision.

Here we present *MindSet: Vision*, a toolbox aimed at facilitating testing DNNs on visual psychological phenomena by addressing all the problems presented above: our main contribution is to provide a large, easily accessible, parameterized, set of 30 image datasets (and related scripts to re-generate and modify them) accounting for a wide array of well-replicated visual experiment and phenomena reported in psychology. Our stimuli cover aspects of low and mid-level vision (including Gestalt phenomena), visual illusions, and object recognition tasks. While our stimuli primarily focus on static vision, reflecting the current emphasis in DNN/human alignment research, we recognize the importance of other areas of vision, such as dynamic processes like motion perception. We hope that our work serves as a solid foundation for further investigation into these complex cognitive processes. We provide a high-level descriptions of the visual phenomena in the main text (Section 2) and more detailed descriptions in the Appendix (A).

To facilitate experimentation across a variety of scenarios, each dataset can be easily regenerated across different configurations (image size, background colour, stroke colour, number of samples, etc.). To address the difficulty in testing DNNs on these stimuli, we provide scripts for using one (or more) of three methods: Similarity Judgment Analysis, Decoder Approach, and Out-of-Distribution classification (Section 3). We provide examples illustrating how to use these scripts with a classic feed-forward CNN (ResNet-152), and an extensively documented code (Section 4).

With *MindSet: Vision*, we aim to bridge the gap between computational modeling and psychological research, bringing experimental studies that manipulate independent variables to the forefront of developing and evaluating of DNN models of human vision. We also hope this initiative will drive further interest in other areas of human psychology, such as memory,

language, and speech perception when attempting tounderstand and replicate human-like intelligence in machines.

## 1.1 RELATED WORK

Several recent studies share some similarities with our project: Roy et al. (2024) propose a human-validated dataset of five types of brightness illusions and benchmark three DNNs on their ability to identify and localize these illusions. Zhang et al. (2023) introduced a new dataset containing five types of visual illusions falling into two categories: color constancy and geometrical illusions. The authors formulated four tasks specifically designed to examine the performance of Visual Language Models, finding low alignment with human responses. Lonnqvist et al. (2024) developed the Good Gestalt datasets, consisting of six types of datasets covering several types of Gestalt grouping principles, including Closure, Continuity, and Proximity, aimed at testing a Latent Noise Segmentation Network. Similarly, Geirhos et al. (2021) developed the model-vs-human benchmark that compares ANN-human classification errors on various 'out-of-distribution' datasets composed of naturalistic images that were modified in various ways, including low-level feature manipulations of contrast and spatial frequency, as well as higher-level manipulations, such as generating silhouettes and sketches of images. Evans et al. (2022) used a dataset of silhouettes, line-drawings, and contours, to investigate robustness to these stimuli in DNNs pretrained on CIFAR-10, and Baker et al. (2018) employed a dataset of line drawings and silhouettes from ImageNet classes to investigate model robustness to local versus global features. In comparison to these works, we present a toolbox to test DNNs on visual psychological effects, investigating not only a much richer set of visual phenomena, but providing the code base to regenerate images in batches, changing the parameters, and testing each one of them on a variety of methods.

## 2 DATASETS

We have included datasets from experiments that characterize a wide range of visual phenomena, ranging from low- to high-level vision. We grouped the datasets (indicated in **bold**) into 3 broad categories (see following Sections) as illustrated in Figure 1. Each dataset comprised multiple sub-conditions designed to test DNN-human similarities, and in some cases, image datasets used to train decoders, as described in Section 3.3.

While most of the stimuli are created by us, in a few instances we incorporate stimuli from external sources (when needed, permission was obtained from the authors). In all cases, the stimuli have been integrated into a versatile framework which offers significant flexibility in adjusting parameters such as image size, background, stroke colour, and more, to allow their application to a variety of models and methodologies. Given the extensive range of datasets provided, we only offer a brief summary for each in the article, and provide more details in Appendix A, including details about the suggested way to test each dataset, and the expected result for model-human perceptual alignment. All resources are open-source and freely available under the MIT license at `https://github.com/MindSetVision/MindSetVision` with additional Kaggle links to datasets we generated provided in the repository README.

## 2.1 LOW AND MID-LEVEL VISION

A fundamental low-level vision phenomena is captured by **Weber's Law** (Weber, 1983), which states that the minimum physical change of a stimulus on some dimension (e.g., its size) that is noticeable to an observer is a constant ratio of the original stimulus value on this dimension. For example, it is equally easy to distinguish between line lengths of 1 and 2 cms and between 2 and 4 cms. We created a dataset that can be used to assess this relation for both line length and stimulus intensity.

Human perception is also sensitive to various **Emergent Features** in which simple image features interact to generate "Gestalts" (Pomerantz & Portillo, 2011a). The dataset is comprised of a set of dots arranged in such a way as to induce the emergent feature of proximity, orientation, and linearity (Pomerantz & Portillo, 2011b; Biscione & Bowers, 2023). Another Gestalt effect is manifest in the **Crowding**/**Uncrowding** phenomenon. In

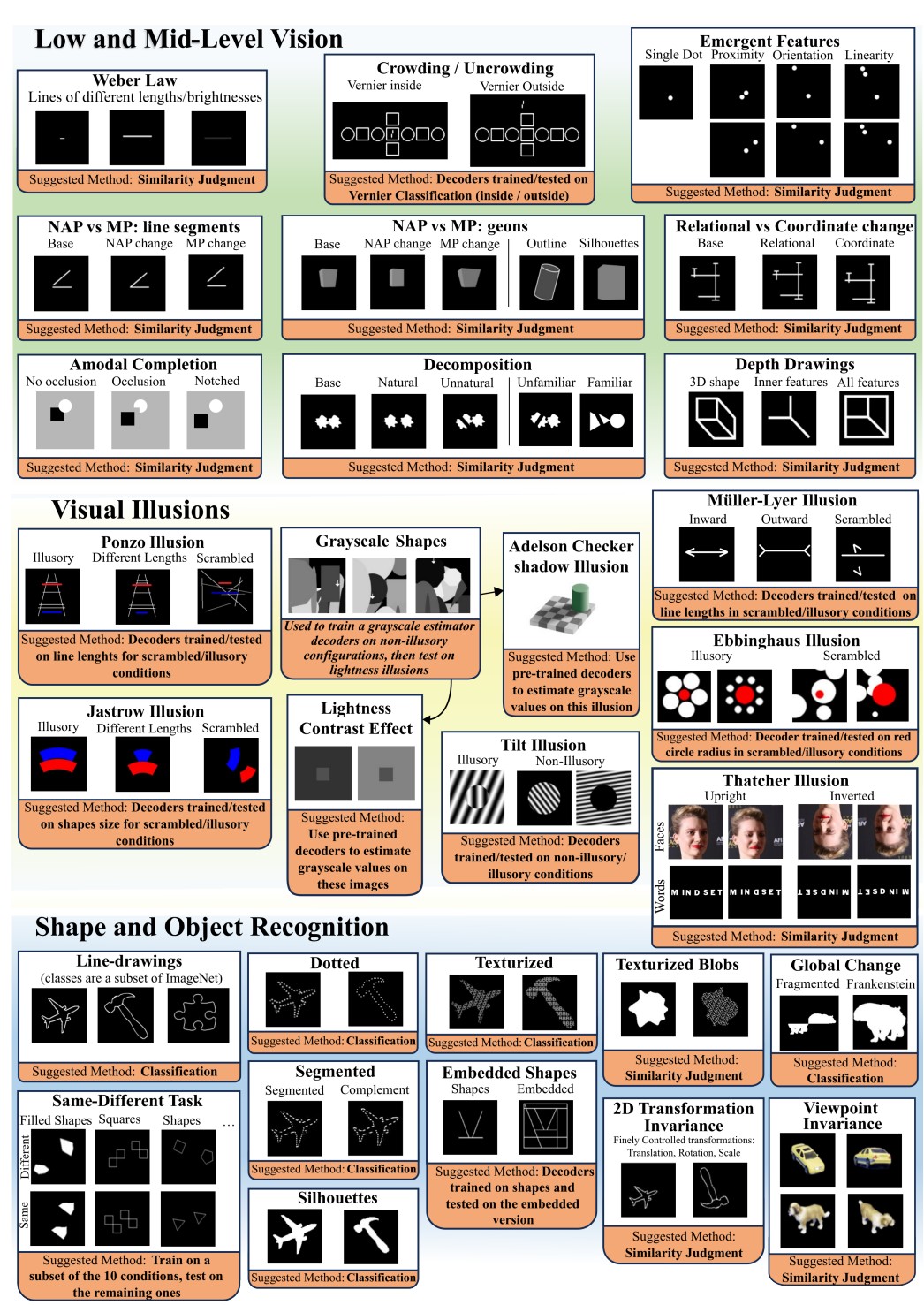

Figure 1: Comprehensive overview of the 'MindSet: Vision' datasets, arranged in three main categories. Each panel represents a distinct dataset, which is further divided into conditions. The images provide examples from these conditions, generated with default parameters.

crowding, the ability to identify an object is compromised by the presence of nearby objects or visual patterns, but in uncrowding, object identification is improved when additional

objects or visual patterns are added to the scene and grouped such that they are segregated from the target. We adapt the dataset from Doerig et al. (2019) so that many crowding conditions with several different shapes can be investigated.

Human perception is highly sensitive to non-accidental image features, that is features that are largely invariant to changes in viewpoint when projected on to the retina (Biederman, 1987), as opposite to accidental features (e.g., degree of curvature) in which the projected image varies with viewpoint. Human vision is known to be more sensitive to changes in images that alter non-accidental compared to accidental properties (Amir et al., 2012; 2014). We use two datasets to examine model sensitivity to these features: one with **3D geons** (Amir et al., 2012) and another with **2D line segments** (based on Kubilius et al., 2017). Similarly, we present a dataset to compare **Relational Changes** with **Coordinate changes** between object parts. DNNs are commonly insensitive to relational change, even after being explicitly trained on these relations (Malhotra et al., 2023), whereas human perception is highly sensitive to relational changes (Hummel & Stankiewicz, 1996).

To identify partly occluded objects, the human visual system groups contours and surfaces through an amodal completion process (Nakayama & Shimojo, 1992). The **Amodal Completion** dataset (based on Rensink & Enns, 1998) enables the investigation of these processes using images with shapes that are either occluded, unoccluded, or "notched". These latter shapes are unoccluded but notched in such a way as to maintain a high degree of feature similarity with their occluded counterparts.

With the **Decomposition** dataset we provide a mean to test the extent to which DNNs group object parts in a human-like fashion. We designed familiar and unfamiliar objects composed of two conjoined parts that undergo what humans would perceive as a "natural" or "unnatural" break (inspired by Jacob et al., 2021). In the same work, Jacob et al. (2021) showed that VGG-16 trained on ImageNet did not possess human-like sensitivity to images that could be interpreted as 3D-shapes by using a set of stimuli based on Enns & Rensink (1991). Accordingly, we reconstructed this **Depth Drawings** dataset.

## 2.2 Visual Illusions

Visual illusions are not mere curiosities, but often arise from adaptive perceptual processes (Gregory, 1997). Detailed computational models of multiple illusions have been advanced providing theoretical insights into the mechanisms that underlie them (e.g., Grossberg, 2017). We provide datasets exploring illusions related to size perception, orientation, and lightness contrast.

Several illusions relate to size perception. In the **Müller-Lyer illusion** (Day & Knuth, 1981), arrow-like segments at the ends of equal-length lines impact our perception of length. In the **Ponzo illusion** (Ponzo, 1910), two equal-length horizontal lines cross a pair of converging lines. In this configuration, the top line looks longer, an illusion often explained as related to the process of inferring depth. In the **Ebbinghaus illusion** (Aglioti et al., 1995), the size of a circle is perceived differently depending on the size of surrounding circles. Similarly, in the **Jastrow illusion** (Jastrow, 1892; Day & Knuth, 1981), a specific arrangement of identical objects affects our perception of their relative size. For all these illusions, we provide both an illusory condition and a condition in which all elements of the original illusion are "scrambled up", so that a decoder can be trained to predict a specific feature (e.g. the size of the centre circle in the Ebbinghaus illusion) and subsequently tested on illusory configurations.

In the **Tilt illusion** (Gibson & Radner, 1937), the orientation of a central grating is perceived as being repulsed from or attracted to the orientation of a surrounding grating. In this case, we provide conditions with either central or background gratings (configurations which do not support the illusion in humans and could be used for training a decoder), and a condition with both (eliciting the illusion in humans) for testing the model. Another orientation illusion is the **Thatcher Effect** (Thompson, 1980). This is a phenomenon where local changes in facial features (like inverted eyes or mouth) are less noticeable when the entire face is upside down, highlighting our sensitivity to orientation in face perception. An interesting unresolved issue is the extent to which this inversion effect is specific to faces (Boutsen et al., 2006; Wong et al., 2010). Together with a dataset of faces and their Thatcherized version,

we also include a dataset of **Thatcherized Words**, that is a dataset of images containing words in which one or more letters are rotated by 180 degrees (Wong et al., 2010).

The **lightness contrast effect** (Von Helmholtz, 1867) and the **Adelson Checker shadow** (Adelson, 2005) illusions reveal how our visual system perceives color and lightness based on context. We provide a **Grayscale Shapes dataset** to train a decoder to output estimates of lightness at a given location of an image (indicated by a small white arrow). After training, the network is presented with test images that induce illusions and help assess whether DNNs show similar effects by pointing the arrow at the relevant parts of the images (see Appendix C.2.6 for a detailed description of this approach).

It is important to note that there is no accepted account for some of the illusions described above. However, even when we have no good understanding of the functional role or the mechanism that drives an illusion, a DNN model of human vision should show similar effect. Indeed, understanding the conditions under which DNNs show an illusion may advance our understanding of why the phenomenon is observed in humans. There are now several articles exploring such illusions in various types of DNNs trained in different ways, with some highlighting similarities (e.g., Benjamin et al., 2019; Storrs et al., 2021; Watanabe et al., 2018) others reporting mixed or discrepant results (e.g., Gomez-Villa et al., 2019; Ward, 2019; Zhang et al., 2023); for a review of the relevant findings, see Kanwisher et al. (2023).

## 2.3 Shape and Object Recognition

DNN object recognition is much more sensitive than human vision to distributional shifts from the training set. For instance, humans can easily identify line drawings the first time they are exposed to them (Hochberg & Brooks, 1962), whereas DNNs perform poorly under these conditions (Evans et al., 2022) and need to be trained on line drawings in order to recognize them at human levels (Singer et al., 2022). We have included the **line drawing** and **silhouettes** datasets (from Baker et al., 2018; Baker & Elder, 2022) and also manipulated them in various ways to construct additional datasets. The line drawings were converted into dotted contours (**Dotted line drawings**), line segments (**Segments line drawings**) (Biederman & Cooper, 1991), or "texturized" (**Texturized line drawings**). The texturized images are composed of oriented lines/characters applied to either/both the background or/and the inside area of the line drawing. In all these cases, the resulting images are easily identifiable by human observers due to various Gestalt rules that organize the image features into boundaries. We also apply the same texturization technique outlined above on unfamiliar "blob"-like shapes (**Texturized Unfamiliar dataset**). Human observers have no difficulty matching a novel "blob" object to its texturized counterpart. In addition, we provide a dataset of fragmented images based on Baker & Elder (2022) in which the global features of silhouettes or line drawing are modified by reflecting the top part of an object along its vertical axis, leaving the local features mostly unchanged (**Global Modifications** dataset). Human performance on these stimuli is greatly reduced but typically DNN performance is largely unchanged, suggesting that human vision is more sensitive to global object structure and DNN vision is more sensitive to local features.

The **Embedded Shapes Dataset** (inspired by de-Wit et al., 2017) provides another condition that greatly impacts on human perception, by embedding geometric shapes within complex arrays of lines in ways that camouflage the original shape. We include both the original images from de-Wit et al. (2017) and a procedurally generated dataset in which random polygons are embedded into a configuration that makes recognition challenging for humans.

The human visual system supports object recognition following a wide variety of transformations (Tanaka, 1996; Blything et al., 2021). Importantly, this extends to cases in which an object has only been viewed at one pose. Previous works suggest a complex link between DNN pretraining and their object recognition capabilities under object transformations (Biscione & Bowers, 2021; 2022). To test whether DNNs share these capacities, we provide a dataset in which translations, plane rotations, and scale changes (**2D Transformations**) are applied to line drawings. To test for **Viewpoint Invariance** (e.g. the ability to recognize an object from a new viewpoint after a rotation in depth) we adapt the ETH-80 dataset, (Chen et al., 2020a) allowing for controlled variation in azimuth and inclination.

Finally, we provide a dataset to test whether DNNs possess the ability to solve a basic form of visual reasoning task, namely, the **Same**/**Different** task. Drawing from Puebla & Bowers (2022), our dataset comprises images composed of pairs of objects, which may be identical or different. These images are organized into ten conditions that vary in their visual form, such as 'filled polygons', 'open squares', and 'colored shapes'. While humans effortlessly accomplish this task across all conditions without training, DNNs often struggle when the training and test images come from different conditions.

## 3 Testing methods.

Each dataset is designed to align with at least one of three methods of testing, but other approaches can be used as well. We discuss further possibilities in Appendix B.

### 3.1 Out-of-Distribution Classification

In this approach, a DNN pretrained on one dataset is tested on a new dataset composed of out-of-distribution images taken from the trained classes (e.g., a DNN pre-trained on ImageNet is tested on line drawings taken from the same categories). This approach is well suited for most of the Shape and Object Recognition datasets that use images from ImageNet categories modified in such a way that human observers have no trouble recognizing them, even without training. We provide scripts to test a wide variety of vision models.

### 3.2 Similarity Judgment Analysis

This method involves assessing the pairwise similarity of activation patterns in DNNs (using a Cosine Similarity or an Euclidean Distance metric) evoked by pairs of images and comparing these similarities to human performance. This method has been used to assess how well DNNs capture human similarity judgments (Peterson et al., 2017) and response times to identify target stimuli from foils (Biscione & Bowers, 2023). It is often useful to carry out these analyses across multiple layers of DNNs given that some psychological phenomena are known to manifest at earlier or later stages of visual processing. A DNN mimicking human perception should show relevant similarity effects at the relevant layers. One key advantage of this approach is that it can be applied to novel images that cannot be classified by a DNN.

To illustrate, we applied this method to the Texturized Unfamiliar dataset (Figure 2). The human visual system groups elements in a scene by texture (Beck et al., 1983) and classify objects by their shapes (Biederman, 1987). Accordingly, texturized versions of the same shape should be judged as more similar than texturized versions of different shapes. To explore if DNNs exhibit similar behaviour, we input pairs of images into a ImageNet pre-trained ResNet-152 and, for each pair, we computed the Euclidean Distance between their internal activations at every processing level. A human-like response is indicated by a smaller distance for pairs of the same compared to different shapes. ResNet-152 exhibited a weak manifestation of this pattern in the early layers, a reduced effect in the later layers, and no effect in the output layer. By contrast, the human visual system supports similarity judgements on the basis of shape-based representations that are computed following the early stages of visual processing.

### 3.3 Decoder Method

In this method a small, often single-layer, "decoder" network is attached to a layer of a frozen DNN and trained on a task designed to reveal how the DNN encodes a specific type of information. For instance, a frozen DNN might be presented with a set of images that contain a target object varying in size, colour, and orientation, and a decoder is trained to output the value of one or more of these properties at a given layer. We provide scripts for both classification and regression training, and scripts to train and test a series of five decoders at varying levels of a ResNet-152 model. Although these scripts are tailored to ResNet-152, they can easily be used as a template to streamline the adaptation of this technique for different networks.

To illustrate, consider the Ebbinghaus Illusion. The Ebbinghaus dataset we provide consists of three conditions: two illusory conditions in which a red centre circle (at different radii) is surrounded by either small or large white circles (flankers) in a configuration that, in humans, induces a biased size estimation of the centre circle: the circle appears larger when surrounded by small flankers, everything else being equal. Another condition again contains a red centre circle of different sizes, but the surrounding circles are placed randomly on the canvas so that they would not elicit any illusion on a human observer. We use the latter condition to train decoders attached to a ImageNet pre-trained ResNet-152 model with frozen weights. The task consists of estimating the size of the centre circle. After training, we feed the illusory images to the decoders. For a network to exhibit the Ebbinghaus visual illusion, the size of the centre circle should be overestimated for small flankers and underestimated for big flankers. We did not find this pattern in ResNet-152 and, indeed, no significant difference across prediction errors for the different conditions was observed (result for one decoder shown in Figure 2).

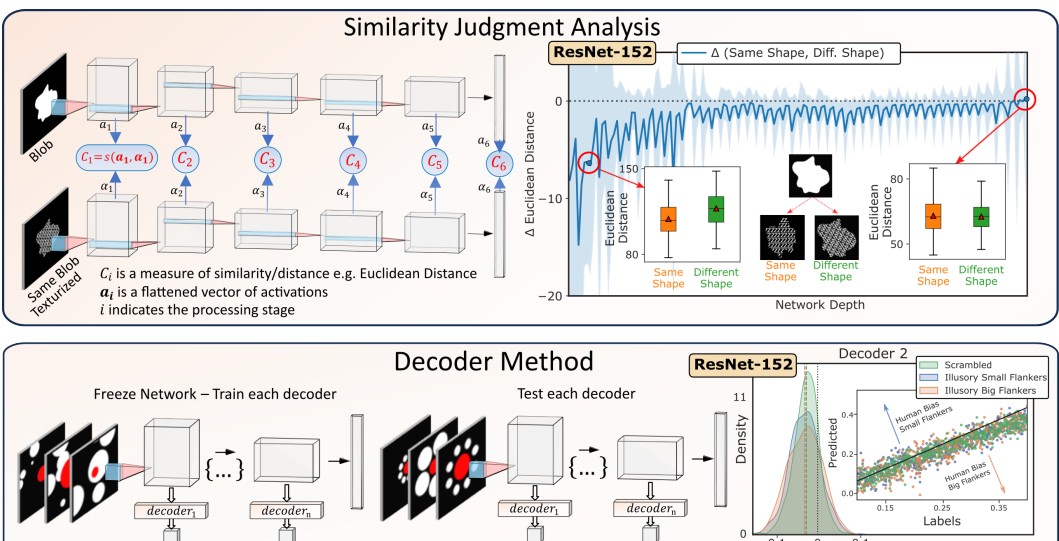

Figure 2: Depiction of two of the three proposed methods of evaluating DNNs in the context of two representative datasets. The first method, out-of-distribution classification, is not depicted here. The Similarity Judgment Analysis (top panel) involves feeding pairs of images to DNNs and comparing the elicited internal representations. We illustrate this method via the 'Texturized Unfamiliar' dataset, showing that the network possesses human-like responses in earlier layers which diminish in the later ones. The Decoder Method (bottom panel) involves training and testing a simple linear layer attached to different stages of a frozen network. In the given example, we assess the response to the Ebbinghaus illusion. Our findings indicate an absence of illusory perception. Both examples use an ImageNet pre-trained ResNet-152.

## 4 CODE AND RESOURCES

We provide both ready-to-use datasets and scripts to generate them with varying parameters. Most of the ready-to-use datasets' size span to around 5,000 images per condition, and larger dataset can easily be generated using the provided scripts. To separate code and configuration, each dataset generation script relies on a configuration file, consisting of a plain-text file in TOML format specifying all the available parameters for that dataset. Some parameters are used across most datasets, such as image size, background colour, and number of samples. Other parameters are dataset-specific, for example the size and distance of dots in the Dotted line drawing dataset. For convenience, the same configuration file can specify the configuration for multiple (or all) datasets, so that they can be generated in batches. The "default" configuration file we used to generate the ready-to-use versions is included,

which can be used as a template. The output of each script is the dataset itself (with several sub-conditions depending on the dataset) together with a CSV annotation file, containing the path and parameters of each generated image.

We also provide the code and utilities to evaluate DNNs using the three methods noted above. Each method is highly configurable through TOML files, with options including the type of data-augmentation to apply, the network architecture, the metric to use for the similarity judgments and more. Users have the flexibility to choose specific factors from this file for analysis, extending beyond the factors that we deemed the most relevant for each task. For example, in the script testing the Ebbinghaus Illusion used for Section 3.3, a decoder is trained to predict the normalized size of the centre circle. However, a different research goal might involve predicting the size of the flankers. This can be achieved by simply specifying the corresponding column ('NormSizeFlankers') in the annotation file, without needing to re-generate the dataset or change the code.

Each method produces pandas DataFrames (McKinney, 2010) as its output, which can be independently analyzed. Additionally, supplementary files containing simple tests and comparisons that serve as a springboard for further and more detailed analysis are automatically generated. Comprehensive documentation for every configurable option across all datasets and methods. Additionally, we offer guidance on the general usage of various scripts and utilities through several examples and multiple README files on the GitHub page.

For licencing details see Appendix D.

## 5    LIMITATIONS

While *MindSet: Vision* offers a valuable resource for exploring visual psychological phenomena using deep neural networks, there are several limitations to consider. Firstly, our focus is primarily on static visual tasks that do not involve high levels of reasoning and are not directly connected with other areas of cognition such as language and memory.

Secondly, the methodology for comparing DNN performance to human participants often allows only for qualitative comparisons, as quantitative comparisons may not be feasible with the current analysis methods.

Thirdly, we did not conduct extensive testing on a range of model architectures. We do cite many articles reporting failures of models to replicate human behaviour, and two of the stimulus conditions included in the toolkit came first and second place in the behavioral benchmark track of Brain-Score competition. The competition awarded benchmarks which produced the poorest overall alignment with human behavior across 21 models[1][2]. But our goal in introducing the toolkit is not to highlight failures of many models nor to provide a single benchmark score that characterizes how human-like a model is overall. Rather, it is to better understand *how* models fail or succeed in manifesting specific visual phenomena, and this will involve extensive experiments using each of the provided datasets. We hope the toolkit inspires these research programs.

Lastly, while we have selected phenomena based on well-replicated and famous visual experiments, there may be additional phenomena that are not covered by our selection, even within the realm of static vision. These limitations underscore the need for further research and development in the field of computational modeling of human vision to address these gaps and enhance the utility of *MindSet: Vision* as a comprehensive toolbox for studying visual perception.

## 6    CONCLUSION

There is much interest in DNNs as models of human vision, but relatively little research is concerned with how DNNs capture key psychological findings. When DNNs are tested against key psychological findings, they often fail (Bowers et al., 2023b). And when they do

---

[1]BrainScore Competition website

[2]YouTube re-stream of the Competition Event at CCN 2024

succeed, it is often because the DNNs have not been severely tested (Bowers et al., 2023a). In our view, to better characterize DNN-human alignment, and to build better DNN models of human vision, it is necessary to systematically test models against key experiments reported in psychology. The MindSet: Vision dataset is designed to facilitate this.

Currently it is quite common to rank models in term of how well they perform across several datasets or tasks. For example, the Brain-Score benchmark (Schrimpf et al., 2018) provides an overall leaderboard that scores any DNN in terms of how good they are at explaining neural activity variance for core object recognition, and the "model-vs-human" benchmark (Geirhos et al., 2021) ranks and scores models in terms of their behavioural overlap with humans in identifying a range of out-of-distribution object datasets. We do not propose to rank models in this way as each experiment in *MindSet: Vision* tests a specific hypothesis regarding how DNNs and humans perceive and encode visual inputs. It makes little sense to provide a score that averages across qualitatively different hypotheses. By making stimuli underlying psychological experiments more accessible, easy to generate, configure, and modify, and by providing ready-to-use scripts to test existing models, we hope that the *MindSet: Vision* toolbox encourages computational modelling researcher to focus on testing their models on key experiments rather than competing on observational datasets that do not support any conclusions regarding the mechanistic similarity of DNNs and brains.

ACKNOWLEDGMENTS

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

APPENDICES

# A  GENERAL DATASET INFO

## A.1  PRE-GENERATED DATASETS

In the pre-generated dataset, we use 224x224 pixel images, and a variable number of samples depending on the task and condition (see section below on the number of samples). However, image size and dataset sizes are parameters that the user can easily modify if needed. The images in the datasets all have 3 channels (RGB). For almost all datasets, the user can specify the background color (either a uniform value, or request a different RGB value for every image), whether to use antialiasing, and the size of the item in the image relative to the whole canvas.

## A.2  DATA AUGMENTATION

We do not apply any affine transformation or other data augmentation techniques during the dataset generation phase. For instance, in the majority of Shape and Object Recognition datasets, a sample typically comprises a modified line drawing or silhouette centrally positioned on a canvas without rotation. We deliberately avoid creating replicas of the same sample with additional transformations. This approach prevents unnecessary expansion of the dataset's size, as most popular deep learning libraries allow for an easy application of data augmentation. Furthermore, our testing methods allow for the application of affine transformation during testing through the configuration file, again avoiding the need to generate pre-augmented datasets.

## A.3  NUMBER OF SAMPLES AND PROCEDURAL GENERATIONS

Some datasets contain a fixed and limited number of samples. For example, the "NAP vs MP: line segments" dataset, recreating the stimuli used in Kubilius et al. (2011), contains 3 conditions with 26 items each. This can be potentially expanded by changing the line stroke, the background color, or by applying dataset augmentation separately. For other datasets much larger samples with extremely low probability of repetition are easily constructed. For example, most Visual Illusions contain a "scrambled" condition in which the elements of the illusions are presented "scrambled up" on the canvas, with varying positions and orientation of each different element. A virtually limitless number of samples can be generated for these conditions, which is important since they are often used for training decoders. The pre-generated dataset typically includes approximately 5,000 samples for these conditions. For users requiring larger sample sizes, they can generate the dataset by changing the configuration argument relative to sample size (e.g., num_samples_scrambled) depending on their needs.

# B  OTHER TESTING METHODS

The datasets featured in MindSet: Vision are suitable for a range of experimental approaches beyond the ones suggested in Section 3. For example, the relational reasoning capabilities needed to solve the Same/Different task could be tested by training a network (which does not need to be pre-trained) on one or a few of the ten available conditions in the Same/Different dataset, and then testing the network on the remaining conditions (as in Puebla & Bowers, 2022)). In a similar vein, the 2D transformations and Viewpoint datasets could be approached by training a network on certain transformations/viewpoints and then testing it on the others, as demonstrated in Biscione & Bowers (2021).

Another efficient method to gain deeper insights into model performance is conducting through ablation: by systematically removing or altering specific components of a neural network, researchers can evaluate the contribution of each specific layers, connections, or whole network components, providing a more nuanced understanding of the network's functioning (Pang et al., 2021; Zeiler & Fergus, 2014; Chen et al., 2020b).

A further method that extends beyond our provided scripts is to input images into Multimodal Large Language Models (LLMs) and query them about what they see (Zhang et al., 2023). Assuming that the language output of the LLM provides a reliable window into its perceptual processes, this approach allows for an interactive examination of the LLM's understanding of the images.

Our preliminary investigations with GPT-4 reveal that while these models are proficient at recognizing silhouettes and line drawings, they struggle with textured representations of familiar objects. For example, a textured image of a banana was misidentified as either a crescent moon or a pair of scissors, and an airplane was mistaken for a butterfly. Moreover, we found instances in which GPT-4 is influenced from images it has previously processed, that is prior exposure to an image can lead the model to incorrectly identify later, differently textured images as the same as the initially viewed object. For instance, when the model is first presented with the silhouette of a banana, followed by a textured depiction of an airplane, it sometimes erroneously classifies the airplane as a banana.

**Prompt:** What object is depicted in this image?

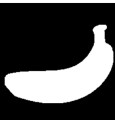

**GPT4:** The object in the image you uploaded appears to be a banana

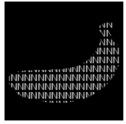

**GPT4:** The object in the image you uploaded appears to be a pair of scissors, composed of ASCII art.

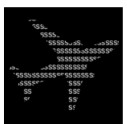

**GPT4:** The object in the image you uploaded is an ASCII art representation of a butterfly.

Figure 3: Samples of GPT4 responses after being prompted by different images from our silhouettes and texturized datasets, each time spawning a new conversation. The model accuracy drops significantly with texturized images.

We ran some preliminary tests on the recently released GPT-4o, finding that this model is much more successful in recognizing textured images, and is able to identify both the overall shape and the individual characters that compose the shape. Relevantly, this model is multimodal in the sense that text and vision are processed within the same neural network, as opposed to GPT-4, in which different networks are used for different modalities.

## C  Detailed Datasets Information

Below we provide more details about the various conditions included in MindSet: Vision and their relevance to understanding human vision. We also highlight the most important parameters for each dataset. For a complete list, refer to the generated HTML page [3] which additionally contains several samples for each condition and each dataset when generated using default parameters.

### C.1  Low and Mid-level vision

There is no sharp dividing line between early and middle visual processing, but early vision extracts low level feature information (i.e., color and luminance contrasts, the orientation of bar and edge segments, and contour segments) from the retinal image. By contrast, mid level vision encodes more abstract aspects of shape, such as surfaces and parts of objects, in an increasingly viewpoint invariant manner. This is where representations of 2D and 3D shapes, material properties, and coherence of the substances and surfaces in the world are computed. That is, mid-level vision builds representations of the distal world from the proximal stimulus. This is an ill-posed problem, and accordingly, various heuristics (such as

---
[3]MindSet: Vision summary HTML

Gestalt perceptual grouping cues) are employed to provide the best estimate of the distal world. Illusions are striking examples of failures to correctly encode the distal world from the proximal stimulus. The experiments we include in MindSet: Vision largely focus on mid-level vision.

### C.1.1 Weber Law

**Psychological Significance.** The Weber Law (or Weber-Fechner Law) quantifies the psychophysical relation between changes in the world and changes in perception. The law states that the minimum physical change of a stimulus on some dimension (e.g., its size) that is perceptible to an observer is a constant ratio of the original stimulus value on this dimension. For example, it is easy to distinguish between line lengths of 1 and 2 cm, but difficult to distinguish between lines of length 100 and 101 cm, despite the fixed difference in length (1 cm). To make the latter distinction equally salient, the two stimuli should be 100 and 200 cm (a fixed ratio of 2). Although Weber's Law breaks down at extreme values, this relationship applies to a wide range of dimensions, from weight, length, size, brightness, and even numbers. Weber's Law reflects the more general observation that perception is often based on relative rather than absolute encoding of stimulus dimensions. Importantly, Weber's Law is often manifest in early visual areas (Hess & Hayes, 1993), and indeed, in some cases, at the level of the retina (Barlow, 1965; Freeman et al., 2010; Treisman, 1964).

Jacob et al. (2021) reported that the convolutional DNN VGG-16 showed a human-like Weber Law effect when encoding line-lengths. However, the authors only observed Weber's Law for line lengths in the late convolutional layers of the network, did not assess whether discrimination was a constant ratio of the original stimulus (they employed a weaker test), and failed to observe a reliable effect for image intensity.

**Dataset.** Images in this dataset are composed of a simple horizontal white line with varying length and brightness values. Configurable parameters include line width, min/max values for length and brightness. To assess DNNs sensitivity to Weber's Law, a similarity judgment analysis assesses whether the relative change in the perception of these stimuli (as measured by the level of unit activation in the inner layers of a pre-trained DNN) adheres to a logarithmic relationship with the stimulus strength (e.g. line length).

### C.1.2 Crowding / Uncrowding

**Psychological Significance.** Our ability to identify objects is impaired by the presence of nearby objects and shapes, a phenomenon called crowding. At the same time, in some conditions, the inclusion of additional surrounding objects makes the identification of the target easier, a phenomenon called uncrowding. This is illustrated in Figure 4, in which participants are asked to perform a vernier discrimination task by deciding whether the top vertical line from a pair of vertical lines is shifted to the left or right. When these lines are surrounded by a square rather than presented by themselves performance is impaired. However, the inclusion of additional squares dramatically improves performance. This is thought to reflect a Gestalt process in which the squares are grouped together and then processed separately from the vernier (Saarela et al., 2009). Standard DNNs are unable to explain uncrowding (Doerig et al., 2020; Francis et al., 2017), but the DNNs inspired by the LAMINART model of Grossberg and colleagues (Raizada & Grossberg, 2001) designed to support grouping processes can capture some aspects of uncrowding.

**Dataset.** Based on Doerig et al. (2020), with code adapted with authors' permission. Images are composed of a 'vernier' stimulus (two parallel line segments with some offset) placed either inside or outside a set of random flankers (squares, circles, hexagons, octagons, stars, diamonds). Each configuration has from 1 to 7 columns and from 1 to 3 rows of flankers with a variety of same/different shape patterns used. The vernier can be left/right oriented. The suggested method for this dataset (as per Doerig et al., 2020) consists of attaching a decoder at several stages of a pre-trained DNN. The decoder is trained and tested on a classification task to discriminate between left/right types of vernier but, significantly, during training, the vernier and the flankers were non-overlapping, whereas during test, the vernier was often placed inside one of the shapes, allowing the measuring of (un)crowding effect

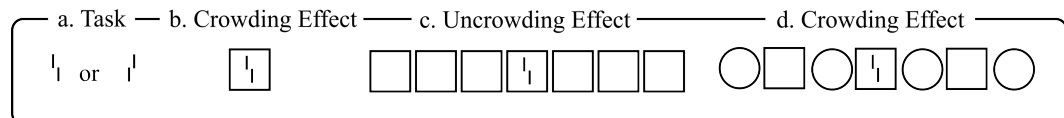

Figure 4: Illustration of the Crowding and Uncrowding effect. a. Observers perform a vernier discrimination task. A standard approach consists of measuring the vernier offset for which observers correctly discriminate in 75% of the trials. With the vernier alone, the offset is quite small. b. When a square is added the performance drastically drops (that is, the threshold-offset increases). This is the classic crowding effect. c. Adding more flankers increases performance again. This is referred to as uncrowding. d. The magnitude of crowding and uncrowding effects is contingent upon both short-range and long-range spatial interactions between visual elements. Furthermore, the specific characteristics and spatial positioning of flanker stimuli play a crucial role in modulating these effects. For example, the performance drops again for the depicted pattern.

through change in classification accuracy across test conditions. A model with human-like visual characteristics should match human perception with regards to both crowding and uncrowding effects, following the pattern in (Doerig et al., 2020; 2019). Users can specify whether the size of the flankers varies or is fixed across samples.

### C.1.3 Emergent features

**Psychological Significance.** Emergent features provide a compelling example of "the whole is different than the sum of its parts". Pomerantz and colleagues (Pomerantz & Portillo, 2011a; Pomerantz et al., 1977) relied on a simple visual search paradigm where participants were asked to identify a target amongst foils. They devised several different types of target and foil stimuli, but the simplest were composed of dot patterns as depicted below. Participants viewed a set of 4 panels, each of which contained a single dot. Three of these panels were identical (dots were in the same location) and one outlier panel (where the dot was in a different location). The task was to identify the outlier panel as quickly as possible. In the single dot condition, the outlier was simply the panel with a dot in a unique position. In the critical emergent feature condition(s), a dot (or more) was added to the single dot images as context. The context dot(s) was in the same location in all panels. Because these added dot(s) were identical in all four panels, there were no new features that could be used to facilitate the identification of the outlier other than configural "emergent" features. For example, in the top row of Figure 5, the extra dot (depicted in the middle column) produces the emergent feature of "orientation", and in the bottom row, the extra dot produces the emergent feature of proximity. The critical finding was that participants could identify the location of the outlier panel more quickly in the emergent compared to the baseline condition. That is, the "whole" was more discriminable than the sum of its parts.

Biscione & Bowers (2023) carried out a series of studies assessing whether DNNs were sensitive to a range of emergent features that facilitated human performance, including testing DNNs on the dot stimuli illustrated in Figure 5. It was observed that DNNs did show some sensitivity to some of the emergent features, but only at the later layers of the network. This is problematic given that these emergent features are thought to be computed relatively early in the visual system, such that they support rapid "pop out" search.

**Dataset.** Adapted from Biscione & Bowers (2023). The dataset consisted of sets of paired images. Each set includes four conditions: a base condition (single dots), and composite conditions (orientation, proximity, and linearity). The 'single dots' condition consists of paired images in which each image contains a single dot placed at a different location. In the composite conditions, one or more dots are added to both images of the base condition, in the same locations, in such a way that it would elicit different emergent properties when combined with the original single dots. In the orientation and proximity conditions, the added dot results in different orientation/proximity features. In the linearity condition (generated by adding a dot to the orientation condition), the added dot would either be placed on

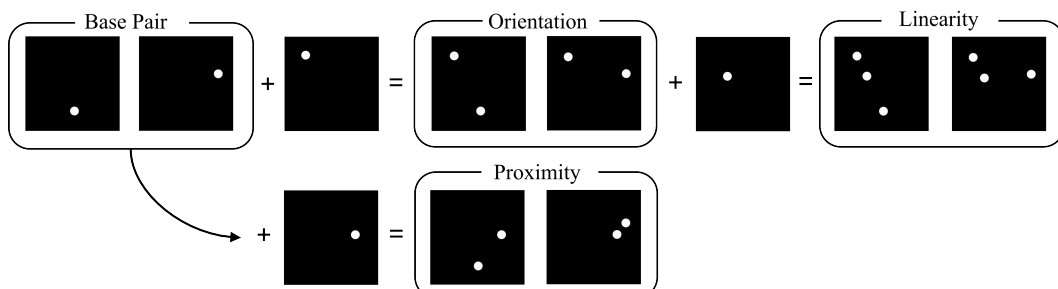

Figure 5: Schematic of the generation procedure for producing a set of dotted stimuli. Starting with a pair of images in which the only discriminant feature is the location of a dot (Base Pair), an additional dot is added, yielding the Emergent Feature of proximity or orientation. The Emergent Feature of linearity is obtained by adding a dot to the orientation pair. Notice that the added dot is the same to both elements of the pair so it does not add on its own any discriminative features, but it generates additional features in relation with the surrounding dots.

a straight line with the other two dots or on a different path. Each dot was constrained to be located at a distance of at least 20 pixels from one another, and 40 pixels from the border. By computing the difference in similarity scores between each composite condition and the base condition, we can compute how much each emergent feature impairs/facilitates distinction of the additional dots. For example, if the average 'orientation' pair is found to be easier to distinguish (through a similarity analysis of the internal activations of the network) than the pairs from the 'single dot' condition, then we can infer that the network is sensitive to orientation (as the additional dot in the orientation condition was not-diagnostic, e.g. the same for both images in each pair). The same comparison with the 'single dot' pairs can be performed for the proximity and linearity conditions. The overall pattern of similarity scores should match human results, in which the highest effect is obtained through the feature of proximity, followed by linearity, and then orientation (Pomerantz & Portillo, 2011a; Biscione & Bowers, 2023).

### C.1.4 Decomposition

**Psychological Significance.** The visual system represents objects in terms of their parts, separating regions at points of deep concavity (Hoffman & Richards, 1984). Perceptually, searching for an object broken into its natural parts among a set of unsegmented versions of the same object is significantly more challenging than locating the same object when it is segmented at points that do not correspond to its natural divisions. In other words, a segmentation at natural points preserves the basic parts which make up the object and therefore make the segmented version more similar to the uncut object when compared to an 'unnatural' segmentation. There is good evidence that this occurs relatively early in visual processing (Xu & Singh, 2002). To assess whether DNNs encode objects into parts in a similar manner, Jacob et al. (2021) compared the internal representations of a base object composed of two parts to two segmentations of the object, one natural and one unnatural. The assumption is that a natural segmentation of the image will be encoded in a more similar way to the whole object (the segmented images maintain the integrity of parts that compose the complete object). However, they reported that the VGG-16 did not show this pattern, suggesting that DNNs do not encode objects by their parts, or at least, not in a way similar to humans.

**Dataset.** The dataset consists of a variation of the images used in Jacob et al. (2021): instead of a single object composed of two parts, we used two objects joined at a single point of contact. There are three 'split' conditions and two 'familiarity' conditions. The 'split' conditions are: 'no split' in which two parts are touching at one point but not overlapping; 'natural split', in which two parts are separated; 'unnatural split' in which the two parts are touching each other as in the 'no split' condition, but one of the parts is 'cut' and separated

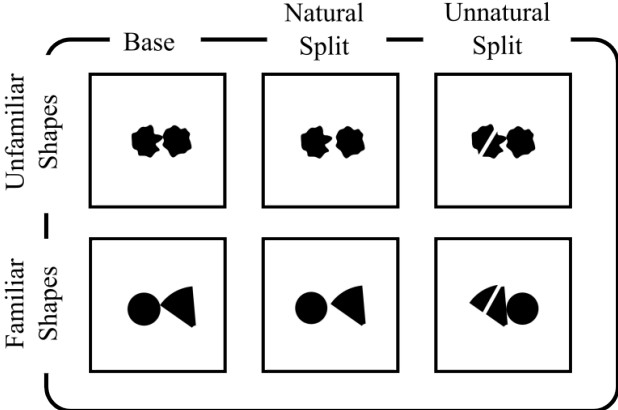

Figure 6: The dataset features base images depicting two objects in contact at a single point. It includes two variations: natural and unnatural splits. In natural splits, the objects are separated, while in unnatural splits, the division occurs within an object itself. Identifying differences between base images and unnatural splits is simpler than distinguishing between base and natural splits. The dataset presents examples with both familiar and unfamiliar shapes, showcasing the diversity in object recognition challenges.

from the rest. The items are silhouettes uniformly coloured on a uniform background, and they can be either familiar or unfamiliar shapes. The familiar shapes consist of the following objects: circle, square, rectangle, triangle, heptagon, and a 50-degree arc segment; the unfamiliar shapes consist of blob-like objects. Within each familiar/unfamiliar condition, all possible combinations of two shapes are used (e.g. a triangle with a rectangle). Configuration parameters include the distance between pieces in the 'unnatural split' and 'natural split' conditions, the colour of items, and the number of different blob-like objects to use for the unfamiliar condition. Following the test from Jacob et al. (2021), similarity judgments between pairs composed of base samples and natural/unnatural splits can be computed for an ImageNet pre-trained network. To match human perception, the natural split samples should have internal representations that are closer to the base samples than the unnatural split samples. This should apply regardless of whether the shapes are familiar or unfamiliar.

### C.1.5 ENCODING RELATIONS BETWEEN OBJECT PARTS

**Psychological Significance.** Humans not only encode objects in terms of their parts, but also the relations between parts which are essential for object recognition (Biederman, 1987). Early evidence for this was reported by Hummel & Stankiewicz (1996) who trained participants to identify a small set of artificial stimuli in which they could easily manipulate relations between parts. Two types of changes were introduced to create foils for the base stimuli. First, a coordinate change in which relations between parts were maintained but the position of a part of the object was changed. And second, a relational change in which there was a categorical change in relations between object parts. They reported that participants were much more likely to mistake foil objects for the base object when the relations between object parts were maintained than when the relations changed (coordinate vs relational change in Figure 7). By contrast, Malhotra et al. (2023) showed that two standard convolutional networks are completely insensitive to these relational features, treating Relational and Coordinate foils equally similar to the Basis objects.

**Dataset.** We recreated images originally contained in Hummel & Stankiewicz (1996), Experiment 5, using white strokes on a uniform background. To compare to human perception, similarity judgments can be computed from pre-trained DNNs by sequentially inputting pairs of images composed of a base shape and either their corresponding coordinate or relational change. A pattern that mirrors human perception would result in greater similarity between the base shapes and their coordinate modifications foils as opposed to their relational change foils.

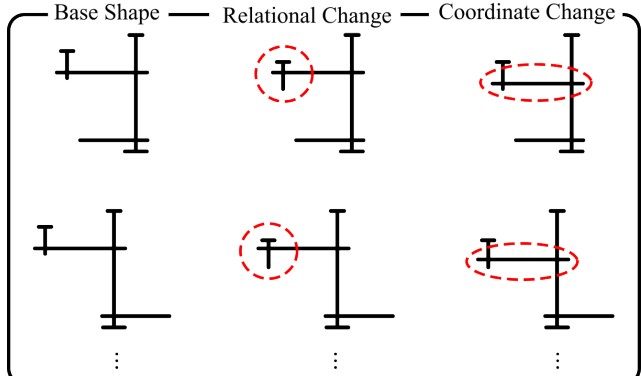

Figure 7: Reproduction of stimuli used in Hummel & Stankiewicz (1996). Starting with a base shape, the Relational Change variant was created by moving one part of the base object up or down (red dashed circle). The move was chosen to change the categorical above/below relation between the circled part and the part to which it is attached. The coordinate change variant was created by moving the whole horizontal (red circled) segment up or down, together with the part moved in the relational change. This resulted in no categorical relations change. Therefore, the perceived difference between a base and its relational-change pair is greater than the perceived difference between the corresponding base-coordinate change pair.

### C.1.6    Encoding of 3D shapes

**Psychological Significance.** The human visual system builds 3D representations of images for the sake of object recognition (Erdogan & Jacobs, 2017; Liu et al., 1995), and some perceptual illusions of size, such as the Ponzo illusion described below, are thought to be a by-product of computing depth information. By contrast, there is little evidence that DNNs infer 3D structure from the 2D images they process. For example, Jacob et al. (2021) tested VGG-16 on three pairs of objects developed by Enns & Rensink (1991): a pair of objects composed of three segmented lines (base pair in Figure 8) are transformed in two different ways (V1 and V2), each time adding the same configuration to both elements of the base pair. Humans were assessed in how quickly they could discriminate the two V1 images and the two V2 images. Discrimination was highly improved for the V2 pair, but not for the V1 pair, most likely the result of enhanced 3D cues in the V2 stimuli.

In contrast, Jacob et al. (2021) obtained no evidence that VGG-16 was better at discriminating the base pair, suggesting a failure to encode their 3D structure.

**Dataset.** We recreated stimuli appearing in Enns & Rensink (1991), using white strokes on a uniform background. Using the similarity judgment method on pre-trained DNNs, a perception akin to humans would result in a significantly lower similarity for the V2 pair compared to both the base and V1 images.

### C.1.7    Amodal completion

**Psychological Significance:** The visual system needs to identify partly occluded objects in the 3D world. A key part of the solution for humans is an amodal completion process in which a surface representation of the occluded object is completed behind the occluder. This process is called amodal because the visual system builds complete surface forms of occluded objects without generating a visible experience of the missing shape. Amodal completion occurs early in the visual system, perhaps as early as V1 (Nakayama & Shimojo, 1992). Various compelling perceptual effects are associated with amodal completion (for a review see Nakayama et al., 1995). Here we include the materials from Rensink & Enns (1998) who showed that humans quickly and automatically encode the shape of partially occluded objects in a visual search task. Amongst the various conditions in their experiments, two illustrate the point most clearly. In the 'Target - Notched square' and 'Target - notched

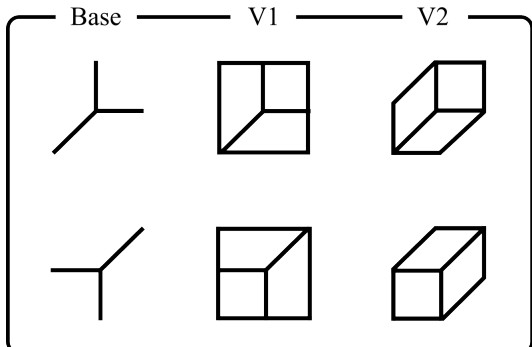

Figure 8: Illustration of the 3D Drawing dataset stimuli. One where segmented lines (Base shapes) are augmented with contextual features to clearly form distinguishable 3D shapes (V2), and another where the additions do not contribute as strongly to depth perception, making shape discrimination challenging (V1). Importantly, the identical contextual features applied to each pair highlight that enhanced discrimination stems solely from the perceived depth, rather than the features themselves.

circle' conditions, participants searched for a notched black square or notched white circle, respectively, among full black square and white circle distractors. None of the objects overlapped in this condition. In the 'Occlusion' condition participants were again searching for notched squares and circles, but in this case notched squares and circles touched to give the impression of occlusion. Search was significantly faster in the 'Notched' condition when compared to the 'Occlusion' condition. This is because in the 'Occlusion' condition the notched squares were perceived as full squares occluded by white disks due to amodal completion. This made the notched black squares much more difficult to find among full black squares. The same was true for the notched white circles in the occlusion condition. This pattern of results suggests that the notched square in the 'Occlusion' condition was encoded as a square early in visual processing (fast visual search is typically characterized as pre-attentive). Jacob et al. (2021) reported that the DNN VGG-16 network pre-trained on ImageNet failed to show any evidence for amodal completion with these stimuli.

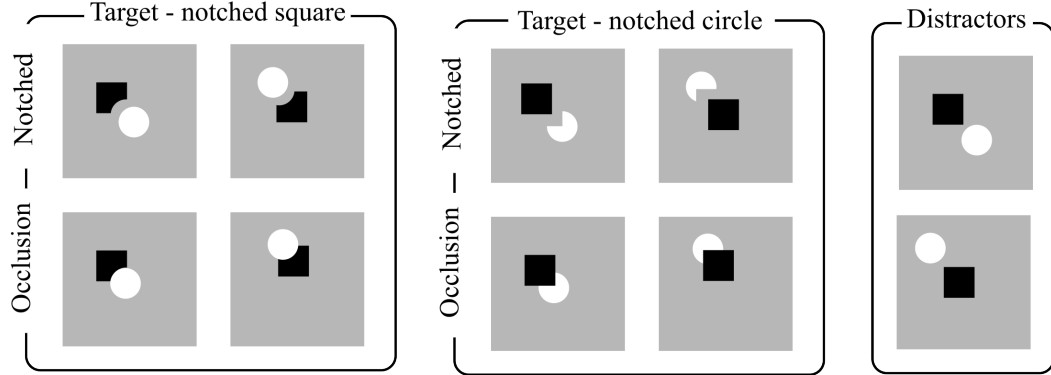

Figure 9: Illustration of the Occluded Shape stimuli as used in Rensink & Enns (1998). The experiment compares three conditions: a baseline condition with squares and disks with no occlusion, an occlusion condition in which one object obscures part of the other, and a notched condition in which the occluded part of the object in occlusion condition is removed. This turns the notch into a distinctive feature and effectively creates a differently perceived shape despite the visible parts being the same as in the occlusion condition. The finding that notched circles and squares are more easily identified than occluded circles and squares is taken to reflect amodal completion.

**Dataset.** We generated samples that look like the stimuli used in Rensink & Enns (1998). We generated samples for the distractors ('unoccluded'), 'occlusion', and 'notched' conditions, with either the square occluding the circle or vice versa. The occluding shape is placed at a variety of degrees from the occluded shape. Each occluded image has a corresponding notched image (that is, using the same shape configurations) so that they can be directly compared. The unoccluded condition is generated by using a non-occluded sample in which the occluding shape is moved radially away from the occluded shape, maintaining the same orientation.

To align with human similarity judgments as measured in Rensink & Enns (1998), a DNN should yield internal activations that result in higher similarity scores for distractor (unoccluded) versus occluded samples (where amodal completion generates representations of the full shape of the notched stimuli), compared to those for (distractor) unoccluded versus notched samples.

### C.1.8 Non-accidental and Metric Properties for Geons and line stimuli

**Psychological Significance:** Human object recognition is highly sensitive to non-accidental properties (NAPs) of an object, that is, visual features that are invariant over rotations in depth. NAPs are hypothesized to be critical for representing object parts such as Geons (Biederman, 1987). For example, curvature (as opposed to a stright line) is a NAP because a curved object in the 3D world will project a curved image on a 2D retina when viewed from most orientations apart from rare "accidental" viewpoints, as when a curve projects a straight contour. NAPs are distinguished from metric properties (MP), features of objects that change continuously with variations over depth orientation when projected on the retina. For example, a curved object in the world will project different degrees of curvature on the retina depending on its orientation to the viewer. A variety of research highlights how human vision is more sensitive to changes in images that alter NAPs (e.g., a change from a curve to straight line) compared to MPs (e.g., changes in degree of curvature) (Biederman, 1987). Kubilius et al. (2017) provided evidence that several DNNs are also more sensitive to image manipulations that alter NAPs, although the effects were most pronounced in later layers of the networks whereas sensitivity to NAP is thought to occur relatively early in human visual processing to encode object parts.

**Datasets.** We have included images of both **2D line segments** based on Kubilius et al. (2017), and **3D Geon stimuli** originally used in Kayaert et al. (2003) to assess the degree in which DNNs are sensitive to NAP vs MP changes. In the case of the Geon stimuli, we have provided a version with shade (as in Kubilius et al., 2011), a version in which no shades are present and the outline is highlighted, and a version in which only the silhouettes are shown, as one concern with the shaded version is that the similarity judgements produced by DNNs may reflect differences in shades as opposed to shape. For each Geon or line segment, a feature dimension (such as the curvature of a Geon) is altered from a singular value (e.g. straight contour with 0 curvature) to two different values (e.g. slightly curved or very curved). The 'reference' condition includes items with the intermediate feature value; in this example, the slightly curved geon. The 'MP change' condition consists of items with a greater non-singular value; in this case, the greater curvature geon. Finally, the 'NAP change' condition includes items with the singular value; the straight contour geon from this example. A human-like similarity judgment would correspond to higher similarity between the reference object to the MP variants than the NAP variants (that is, NAP changes are easier to discriminate). Kubilius et al. (2017) provides a more detailed description of human performance through reaction times that can be directly compared to similarity judgments in DNNs (where higher reaction times correspond to lower similarity).

### C.2 Visual Illusions

There is now a growing number of articles exploring various illusions in various different types of DNNs trained in different ways. Some of these highlight similarities with human perception (e.g., Benjamin et al., 2019; Storrs et al., 2021; Watanabe et al., 2018) while others report mixed or discrepant results (e.g., Gomez-Villa et al., 2019; Gomez-Villa et al., 2020; Ward, 2019; Zhang et al., 2023). For a review of various findings see Kanwisher et al. (2023). The

conditions in which DNNs 'experience' human-like illusions may provide insights into why humans experience these phenomena. For instance, Gomez-Villa et al. (2019) found that various brightness and color illusions can be induced in CNNs trained for image denoising, image deblurring, and computational color constancy. They argue that these illusions are a byproduct of biological processes designed to improve efficiency of low-level visual processes. In addition, (Storrs et al., 2021) found that unsupervised (as opposed to supervised) learning led DNNs to factorize images into encoding of reflectance and illumination that resulted in a human-like perceptual illusion of gloss. Here we consider several classic size, lightness, and orientation illusions. For most of the Visual Illusions Datasets, we provide both an "illusory" condition and a corresponding "non-illusory" condition. The non-illusory condition contains the same basic elements as the illusory one, but they are arranged in such a way that the illusion is not elicited. Both conditions feature procedurally generated images. In the illusory conditions, the stimuli are systematically varied to maintain the presence of the illusion. For example, in the Ponzo illusion dataset, we manipulate factors such as the number of horizontal "railway" lines, the degree of convergence of the vertical lines, and the length of each line. However, we carefully constrain these variations to ensure that the illusion remains effective. We encourage future researchers who wish to modify our scripts or use non-default parameters to regenerate the datasets to verify that the illusory stimuli continue to produce the intended perceptual effects.

### C.2.1 MÜLLER-LYER ILLUSION

**Psychological Significance.** The Müller-Lyer illusion is perhaps the most famous of all illusions. There is no agreed-upon explanation of the effect but the fact that it is observed across species, including fish Sovrano et al. (2016), suggests that it reflects something basic about the architecture of the visual system rather than the training environment. Ward (2019) reported that VGG-19 showed the illusion (to a rough approximation), although they only reported the effect at the final stage of the network. A similar result was reported by Zhang et al. (2022) who reported more robust effects in the higher levels of VGG-19 and ResNet-101.

**Dataset.** The Müller-Lyer illusion stimuli were generated in one of two 'illusory' configurations (with inward or outward 'fins') or in a 'scrambled' configuration. In the latter, the fins are arranged randomly in the canvas, separated from the line segment. In all three conditions, we vary the line length, the position of the line, and the angle of the fins. A method to test whether a DNN is susceptible to this illusion involves training a set of decoders to predict the line length in the scrambled condition set. These decoders are then tested on the illusory conditions. A human-like response would be evidenced by a consistent pattern of both overestimating the line length in the outward illusory condition and underestimating it in the inward illusory condition. Additionally, the illusory effect should be larger for more acute fin angles.

### C.2.2 PONZO ILLUSION

**Psychological Significance.** In this classic illusion, two identical horizontal lines cross a pair of converging lines, a configuration similar to railway tracks. In this configuration, the top line looks longer. The standard explanation is that the visual system assumes that the converging lines are receding in depth and that the upper horizontal line is further away. Given that the two lines project the same length on the retina, the visual system assumes that the upper line must be longer. That is, the illusion is a by-product of the visual system attempting to compute size constancy. Interestingly, there is good evidence that the Ponzo illusion (He et al., 2015), and related size illusions (Sperandio et al., 2012), alter the activation in V1, although this may reflect top-down activation from higher-level visual areas (Chen et al., 2019). Ward (2019) failed to observe a Ponzo effect in VGG-19.

**Dataset.** Two target lines (red and blue) are placed across a railway track pattern. In the illusory condition, the target lines have the same length (varying across samples). In the scrambled condition, the target lines have different length, are still placed horizontally one on top of the other, but all the other segments are randomly placed across the canvas. We include a third condition in which the railway track pattern is used with target lines which

differ in length. The railway track pattern for the illusory and different lengths conditions is composed of converging segments (with a varying degree of convergence), and horizontal segments (randomly placed at different horizontal positions). For all three conditions, the user can specify the number of horizontal segments to use.

The suggested way to test whether DNNs perceive the Ponzo Illusion consists of training a set of decoders on the scrambled condition, to predict either the length of the target lines or a function of the length (for example, the difference between the top and the bottom line lengths). Then the decoders can be tested on the illusory condition. The different lengths condition could be used as a further way of analysing the decoders response. To match human perception, a decoder should overestimate the length of the top target line (or underestimate the length of the bottom line, or output a positive difference in top minus bottom line length, depending on the training setup) in the illusory condition (where the two target lines have the same length).

### C.2.3 Ebbinghaus (or Titchener) illusion.

**Psychological Significance:** In this classic illusion, the perceived size of a central circle is altered by the size of surrounding circles. There is evidence that the illusion distorts the perception of size but not action (Aglioti et al., 1995; Whitwell et al., 2022; but see Franz et al., 2005) and there is evidence that this illusion is mediated by relatively low-level (preattentive) vision (Busch & Müller, 2004). Again, there are different explanations for the phenomena (Rashal et al., 2020). Ward (2019) failed to observe this effect in VGG-19.

**Dataset.** A red target circle is surrounded by a fixed number of white circles (flankers) on a uniform background. In the two illusory conditions ('big' and 'small' flankers) the flankers surround the target circle, and they all have the same size within each sample. In the scrambled condition the target circle is placed in the center, but white circles with random sizes are randomly placed on the canvas. Across illusory samples, we varied the radii of the flankers, the radius of the target circle, the displacement of the flankers around the target. To measure illusory effects in DNNs, decoders can be trained on estimating the circle size or radius in the scrambled condition, and tested on the big/small flankers condition. Human-like perception should induce overestimating in the small flankers condition and under-estimating in the big flankers condition (see example in Figure 2).

### C.2.4 Jastrow Illusion

**Psychological Significance:** In the Jastrow Illusion (Jastrow, 1892), two identical-sized curved segments are perceived as different sizes when one is placed above the other in certain configurations. There are multiple explanations for the phenomenon, but perhaps the simplest explanation is that it is a form of a contrast effect. The length of the concave edge of the upper object in a Jastrow configuration is much shorter than the convex edge of the bottom object, and this contrast drives the perception of size when the edges are closely aligned (Rock, 1975). Rhesus monkeys do not appear to be affected (Agrillo et al., 2019), nor do humans when assessed on grasping behavior (Ozana & Ganel, 2020). As far as we are aware, no one has reported whether DNNs show a similar pattern.

**Dataset.** We used a red and a blue arc shape, either one on top of the other at the centre of the canvas ('illusory' and 'different lengths' conditions) or randomly placed in the canvas with a random orientation ('scrambled' condition). In the scrambled and different lengths conditions the two shapes have different sizes. The size is the same (thus eliciting the illusion) in the illusory condition. To estimate DNNs susceptibility to the illusion the same approach as the Ponzo Illusion can be used.

### C.2.5 Tilt illusion

**Psychological Significance:** In the tilt illusion, a central grating's orientation is perceived as being repulsed from or attracted to the orientation of a surrounding grating. A wide variety of mechanistic accounts of the illusion have been proposed (for a review see Clifford, 2014), and it is argued to be an adaptive feature rather than a bug of a visual system optimized for contour detection (Serre et al., 2020). There is evidence that the illusion

reflects processes in V1 Song & Rees (2018). Linsley et al. (2020) reported that a recurrent DNN optimized for contour detection produces a tilt illusion.

**Dataset.** We provide one illusory condition, in which an oriented grating pattern is presented within a circular mask ('center grating') and a differently oriented grating is placed as the background ('context' grating); and two non-illusory conditions: one in which the background is uniformly colored and only a center mask contains the oriented grating pattern; and vice versa. The samples are varied in their orientation and spatial frequency of the gratings, and in the size of the central grating. Our suggested approach to test whether a DNN perceives the tilt illusion is to train a decoder to estimate the orientation of the center grating, and test it on the illusory condition to check whether the presence of a context affects performance. In particular, the decoder should present the largest repulsive bias at around 20° and an attractive bias at around 70°-80°. Plus, the attractive effect should be much smaller than the repelling effect, and larger for matching center-surround gratings spatial frequencies. (Westhhmer, 1990).

### C.2.6  LIGHTNESS ILLUSIONS

Lightness refers to our perception of the reflective surface of an object (a stable property of an object) whereas brightness is a measure of the amount of light reflected from an object, something that is affected by both reflectance as well as the lighting source. We include two famous illusions related to lightness: the Lightness Contrast Illusion and the Adelson Checker Shadow Illusion. However, to facilitate testing for these and other lightness-related effects, we created an additional dataset called **'Grayscale shapes'**. The purpose of this dataset is not to elicit any illusion in humans or in DNNs but to train a network (or, with our suggested method, a decoder attached to a network) to output the grayscale value of a target pixel.

**Grayscale shapes Dataset.** Each image is composed of 20 overlapping items amongst the following types of shapes (circle, circle sector, circle segment, ellipse, rectangle with straight and rounded corners, heptagon, irregular polygon composed of a random number of edges from 3 to 10). Position, dimension, orientation, and grayscale colour value are randomized for each shape. We place 20 items to be sure that most space in the canvas is filled by an item, but that only a few of them are fully visible. This results in a chaotic canvas with many different shapes with varying grayscale colours but with coherent patterns (as opposed to, for example, having each pixel of a different random grayscale value). In order to target a specific pixel to be predicted by the decoder, a small white vertical arrow (the 'marker') of fixed size is placed randomly on the canvas. The arrow points to the pixel whose value can be used for prediction. Notice that while the images are commonly normalized from -1 to 1 before being fed into the network, the targeted pixel value to predict is in the 0-255 range. Once a trained decoder reaches the desired level of accuracy, it can be tested on other configurations by simply adding the white arrow 'marker' into any image. We call this network with the decoder attached the **color-picker**. We can then test whether an illusory configuration impacts performance of the color picker by placing a white arrow marker at several points of the illusory image and check whether the output is biased in a human-like fashion. This is the approach we use in the Lightness Contrast Effect and Adelson Checker Shadow Illusion.

### C.2.7  LIGHTNESS CONTRAST EFFECTS

**Psychological Significance:** In the Lightness Contrast effects our perception of two identical central gray patches is altered by their surround, such that a patch surrounded by a dark background is perceived as lighter, and the patch surrounded by a light background is perceived as darker. The standard explanation of this is that lightness perception is the product of the relative brightness of surfaces across a boundary given that this ratio will remain constant regardless of the general illumination, allowing lightness (and color) constancy. However, in the lightness contrast context, the mechanism designed to produce lightness constancy results in the central grey squares being perceived differently. These computations are thought to occur in the primary visual cortex (MacEvoy & Paradiso, 2001). Some DNNs can achieve color constancy (Flachot et al., 2022) and other forms of

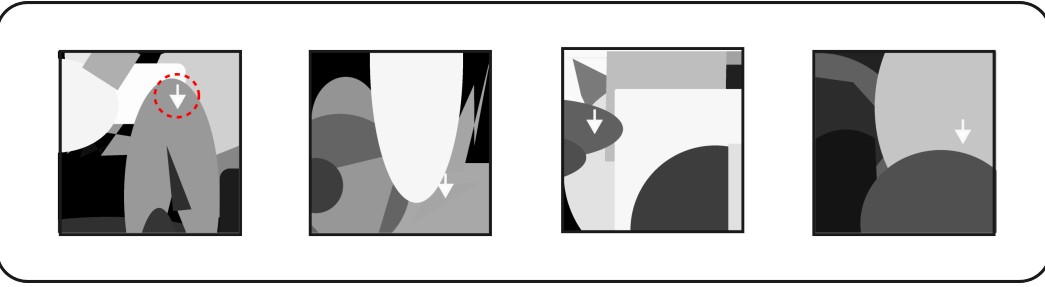

Figure 10: Samples of five images from the grayscale dataset, used to train a color-picker, as detailed in the text.

constancy (Storrs et al., 2021) under some conditions, although there remain questions as to whether this is achieved in a human-like manner. There are also computational theories of the lightness contrast effect (Grossberg, 1983), but we are not aware of any demonstrations that DNNs support this effect.

**Dataset.** The dataset consists of the standard Lightness Contrast configuration: square within a uniform canvas of different grayscale values. The user can specify the grayscale value of the center square, which is kept fixed, while the value of the background is varied. Importantly, each sample is replicated many times with the white arrow marker placed at different locations in the canvas. A color picker network can then be queried for the grayscale value at different locations in order to measure whether the perceived value of the central square is affected by its surroundings.

### C.2.8 ADELSON CHECKER SHADOW ILLUSION

**Psychological Significance:** In this classic illusion (Adelson, 2005), two squares on a checkerboard are perceived to have different reflectance due to one being in shadow and the other in light, despite being the same brightness. This phenomenon is explained with the ability of the human brain to perceive reflectance of a surface as invariant under variation of brightness. In this illusion, the patches inside and outside the shadow reflect the same brightness, and accordingly, the visual system assumes the patch in the shadow must be lighter.

**Dataset.** This dataset simply consists of the Adelson Checker Shadow illusory image replicated many times, grayscaled, with a white arrow systematically placed at different locations of the canvas, covering the whole checkerboard. A color-picker network (that is a decoder trained to predict the value of a marked pixel, e.g. trained on the Grayscale shape dataset, see Appendix C.2.6) is queried at all locations. Critically, the color-picker will show illusory perception if the pixels in the two target patches are seen as two different colours. In particular the pixels of the unshaded patch should be seen as darker than the shaded patch by the network.

### C.2.9 THATCHER ILLUSION

In this illusion the eyes and mouth of upright faces are inverted to produce a grotesque image of a person. The distinction between a normal face and a distorted face is highly salient. However, when faces are upside down, the distortions are much less salient. This effect was originally reported on images of Margret Thatcher, thus the name of the illusion. This effect is sometimes claimed to be more dramatic for faces compared to other categories, lending support to the hypothesis that face processing is special (Boutsen et al., 2006). Other researchers claim that similar effects are observed for other types of objects (Wong et al., 2010). Jacob et al. (2021) demonstrated that CNNs trained to identify faces exhibit a Thatcher-like effect, though they did not assess whether this effect extends to other object categories. The extent to which this effect is specific to faces or can be generalized to non-face objects remains a subject of debate (e.g., see Wong et al., 2010; Bertin & Bhatt, 2004; Dahl et al., 2010). To test DNN sensitivity to the Thatcher Effect for both faces and non-face

dataset, we provide both a Thatcherized dataset of faces and a Thatcherized dataset of words (in which individual letters are rotated).

**Face Dataset.** We provide a small dataset of celebrity faces using a subset of CelebA[4], but the user can specify any folder containing images of faces. Each image is resized according to parameters specified by the user and then reoriented into both an upright and a 180-degree inverted configuration. Furthermore, it is either 'Thatcherized' or remains unaltered. To 'Thatcherize' an image we compute landmarks of the eyes and mouth, compute the bounding rectangle for each, and rotate them around their centre of mass. Blurring on the edge is applied to minimize artefacts. To assess the susceptibility of DNNs to the Thatcher effect in faces, we propose a similarity judgment analysis. This involves comparing the perceived similarity between each upright face and its Thatcherized counterpart, as well as each inverted face with its Thatcherized version. To align with human perception, the latter comparison is expected to yield a higher similarity score than the former.

**Word Dataset.** We employ a collection of 1000 English words or artificially generated sequences of random letters. All entries are uniformly presented in uppercase, covering a range from 3 to 8 letters in length. Following Wong et al. (2010), to simulate the Thatcher Effect for words, we rotate one or more letters by 180 degrees. To increase variability, each word is displayed in one of ten different fonts, with variable font sizes, and includes jitter for each letter. The configurable parameters include the number of words, the exact or range of letter counts per word, the number or range of letters to be rotated, the font size, the level of jitter, and whether to use random strings or English words.

## C.3    Shape and object recognition

A key feature of human vision is that we identify objects largely based on their shape. For example, we can easily identify line drawings of objects with no colour and texture (Biederman & Ju, 1988). To measure shape bias in DNNs there is now a benchmark that tests models on "style transfer" images composed of the shape of one category and the texture of another (Geirhos et al., 2019). Many DNNs, including DNNs that perform at the top of the leaderboard on Brain-Score, rely primarily on non-shape features, as they classify the images based on their texture rather than shape. More recent DNNs trained on much larger datasets have started to show a more human-like shape bias (Dehghani et al., 2023), but there are many additional attributes of human shape perception that need to be accounted for by any DNN model of human vision.

### C.3.1    Identifying line drawings, dotted line drawings, silhouettes, and image segments

**Psychological Significance:** Humans can often identify line drawings of objects as quickly and accurately as photographs, highlighting the importance of shape for object identification (Biederman & Ju, 1988). Interestingly, a child who had never previously been exposed to line drawings can readily identify them, showing that there is no need to be trained on line drawings to identify them (Hochberg & Brooks, 1962). By contrast, DNNs need to be trained on line drawings in order to recognize them at human levels (Singer et al., 2022). Similarly, humans can easily identify silhouettes of objects, whereas DNNs again struggle (although interestingly, they do better with silhouettes compared to line drawings; Baker et al., 2018).

In addition, by exploiting the Gestalt principle of good continuation, humans can recognise line drawings with modified local features when the global shape is left intact. In our experiments, we modified line drawings in three ways: by replacing the continuous line with dots; by replacing continuous lines with segments; and by texturizing them. These images are easily identifiable, and accordingly, DNNs should be able to identify these out-of-distribution images.

**Line drawings dataset**. We use the line-drawing stimuli from Baker et al. (2018), consisting of 36 classes from ImageNet (one line-drawing per class). The line-drawings are white stroke on a uniform canvas (black by default). We used this dataset to build the **Dotted line**

---

[4]https://mmlab.ie.cuhk.edu.hk/projects/CelebA.html

**drawings and Image Segments datasets**. In the former case, the user can specify the dot size and the distance between dots. In the latter case, we have generated complementary images that have complementary segments removed (see Figure 11). That is, each line segment in one image is absent in the other, and together the image is complete. These stimuli are generated by overlapping a grid on the line drawing and deleting complementary sections. The user can specify the grid orientation, the distance between each grid row and column, and thickness of each cell. Participants find these images trivial to identify, and accordingly, DNNs should also. Importantly, humans find complementary images like these hard to distinguish, and indeed, complementary images produce equivalent priming to repeated images, highlighting how the visual system treats them as equivalent (Biederman, 1987). This would also be the case if complementary dots were removed for the dotted line drawings. Thus a second approach to compare humans to DNNs is through a similarity judgment analysis across complementary images, which should return very high similarity value in some hidden layers of DNN.

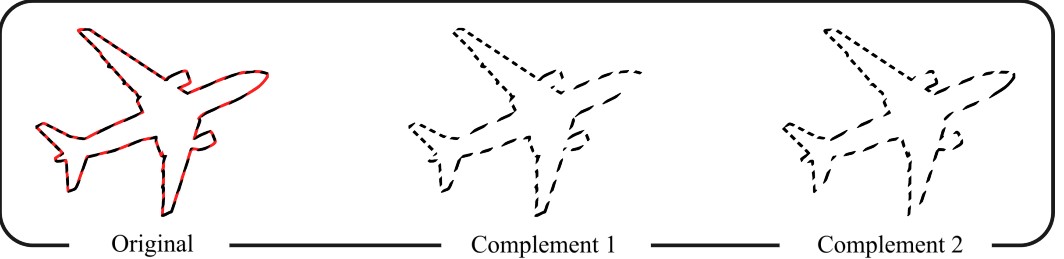

Figure 11: Example of the complementary segment dataset. Each linedrawing results in two images with complementary segments removed. The resulting samples are very difficult to discriminate for humans (Biederman, 1987).

There are many datasets of line drawings available, and the user can specify any folder containing line drawings, to generate a dotted/segmented dataset. The line drawings are expected to be composed of a black stroke on a white background, but can otherwise be of any shape, and the line drawing folder is expected to contain sub-folders for each class (e.g. 'airplane') which can contain multiple line drawing instances for that class (this follows the standard structure used in many deep learning libraries for image classification, e.g. the ImageFolder dataset in PyTorch). Our script will automatically convert the image to a white stroke over a uniform background (black by default).

**Silhouette dataset**. For the Silhouette dataset, we use samples from Baker & Elder (2022); 9 classes from ImageNet, each containing 40 samples. As before, the user can specify any folder containing silhouettes. Alternatively, the user can also specify a folder containing linedrawings (following the same constraints as above), which will be converted into silhouettes. Again, humans find these images easy to identify, so models should as well.

### C.3.2 Identifying familiar and unfamiliar images defined by texture boundaries

**Psychological Significance:** The human visual system can group elements in scenes based on texture, with texture regions defined by the similarity of their elements. This is an example of a Gestalt principle (Similarity) contributing to object recognition (Beck et al., 1983). One way this manifests is through the ability to identify familiar (**texturized objects**) and perceive unfamiliar (**texturized unfamiliar**) by their texture.

**Datasets: Familiar and Unfamiliar Texturized Objects.** We provide a dataset of familiar texturized objects by using line drawings from Baker et al. (2018) as base items. For unfamiliar shapes, we generated silhouettes of blob-like objects. For the pre-generated datasets, the texturization consists of masking the internal contour of a line drawing/silhouette with a pattern of a repeated character with a randomized font size, rotated by a random degree. The character is randomly selected from letters, digits, or punctuation, and we kept the background uniformly colored. When generating images, the user can also specify the

texturization of the background as well, although we have found that doing so will turn object recognition from trivial to challenging, depending on the selected character.

The same approach is used for the unfamiliar shapes. In this case, the user can specify the number of blobs to generate and texturize. For both familiar and unfamiliar datasets, the user can specify how many texturization samples to generate for each input image.

To measure alignment with human visual perception, the different datasets require a different approach. For familiar shapes, DNNs can be tested by simply assessing classification accuracy. For unfamiliar shapes, a similarity analysis can be carried out. For example, a DNN should find a blob and its texturized counterpart more similar than a blob and a differently texturized blob (see example in Figure 2).

### C.3.3 IDENTIFYING EMBEDDED SHAPES

**Psychological Significance:** The Embedded Figures Test (EFT, de-Wit et al., 2017) is a widely utilized tool in research exploring individual differences in perception, with a particular emphasis on studies of autism spectrum disorder, and as a measure of local versus global perceptual style (Panton et al., 2016). Subsequently, de-Wit et al. (2017) developed a set of stimuli in which several Gestalt grouping principles were manipulated in order to create increasingly difficult matching to sample tasks. They found that the principle of good continuation (operationalized in terms of the number of continued lines from the original shape) impacted performance the most. Each target shape was integrated into four distinct contexts, each exhibiting a progressive increase in the number of lines extending from the target shape into its surroundings. The higher the number of lines extending the shape, the lower the performance, highlighting human susceptibility to camouflage and the role of Gestalt organisation principles in camouflage.

**Dataset.** We used the dataset from de-Wit et al. (2017) who developed simple stimuli in which background lines camouflaged geometric shapes to various extent (Figure 12). Importantly, different embeddings have different levels of continued lines from the original shape, which strongly affects human performance. Furthermore, we developed our version by generating 5 irregular polygons, embedding them in a set of lines, some random and some extending directly from the polygon's edges (similarly to the original dataset). Many camouflaging samples can then be procedurally generated from each polygon. Training decoders to classify the simple geometric forms provides one way to assess the impact of embedding shapes on DNNs. Decoders would be trained on simple shapes (either our irregular polygons or the original shapes from de-Wit et al., 2017) and would then be tested on the embedded version. A DNN with a human-like perceptual system should show reduced ability to identify the shapes, with the level of impairment being a function of the amount of lines originating from the polygon (as in de-Wit et al., 2017). Notice in this case, human alignment requires a degradation of performance after image alteration.

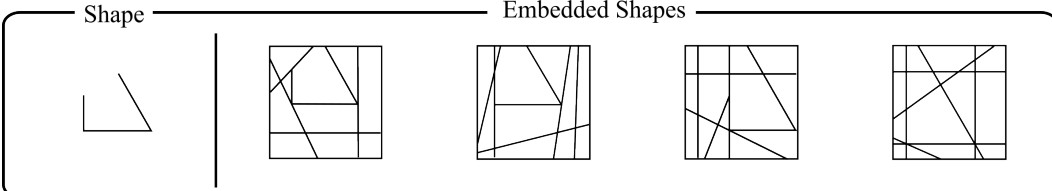

Figure 12: Illustration of one set of items from the embedded shape dataset. These are the stimuli recreated based on Torfs et al. (2014). A basic shape is camouflaged using a variety of horizontal and vertical line, and extending the segments composing the shape. We also provide a variation of this dataset in which, given a set of polygons, camouflaged versions are procedurally generated.

### C.3.4 Sensitivity to Global Shapes

**Psychological Significance:** Human object recognition relies more heavily on global shape representations than on local features, whereas there is evidence that DNNs rely more heavily on local features (Baker et al., 2018), even when trained to have a shape bias (Baker & Elder, 2022). In Baker & Elder (2022), humans and DNNs were presented with silhouette stimuli in their normal format, fragmented, or in a 'Frankenstein' format where most of the local features are preserved but the overall configuration of the image was distorted. That is, the authors modified global shape while maintaining most of the local features. In particular, the 'fragmented' condition (see Figure 13) divides the shape into two distinct, yet adjacent, entities while preserving the local characteristics of the object. The "Frankenstein" scenario involves adjusting the upper section back into alignment with the lower half, so that the bottom and top halves are mirror reversed. This method keeps the object intact as a single entity. Human performance was much reduced in both the Fragmented and Frankenstein conditions, but DNNs performed similarly in the Whole and Frankenstein conditions, highlighting the importance of the local features and the lack of weighting for more global features in driving their performance. Attempts to train networks to focus on the more global aspects of the images failed.

**Dataset.** We provide both the dataset extracted directly from Baker & Elder (2022) and a a version in which the fragmented and Frankenstein versions are automatically generated from any silhouettes or line drawing samples. The Baker & Elder (2022) dataset contains 9 classes from ImageNet, each containing 40 samples. A network with visual capabilities aligned with a humans' should suffer from performance degradation in both fragmented and Frankenstein condition, which can be measured through classification accuracy (as usual, with a network pretrained on ImageNet or some other image dataset).

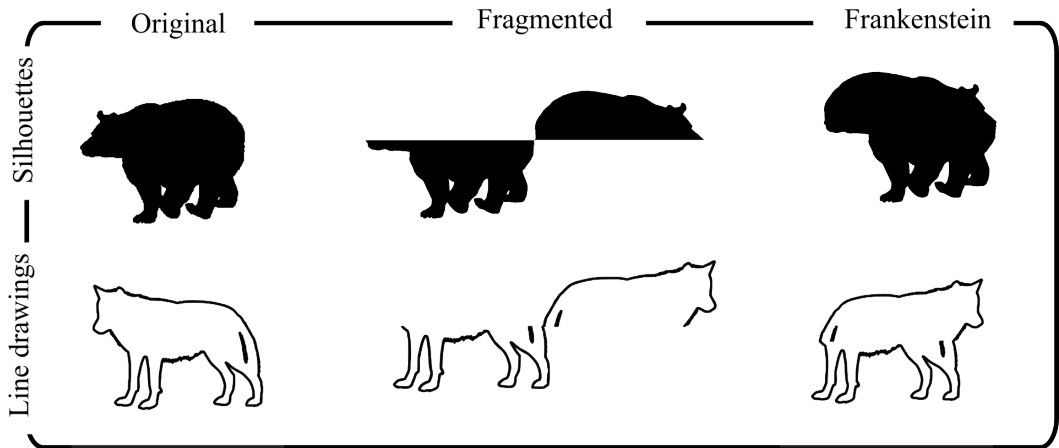

Figure 13: Example of a stimulus and its transformed version, following Baker & Elder (2022).

### C.3.5 Invariance to Object Transformation

**Psychological Significance.** Humans possess the remarkable ability to recognize objects despite the different retinal images the objects project depending on changes in size, orientation, lighting, and placement (Tanaka, 1996). This is performed on-line: an object, once seen in a new or altered form, can typically be recognized instantly in subsequent exposures at different angles, without further training. This capability applies irrespective of the object's familiarity (Blything et al., 2021; 2020). Previous work (Biscione & Bowers, 2021; 2022) found that none of the 7 tested classic visual DNNs possess on-line invariance architecturally (that is: training an object at a viewpoint would not automatically support object recognition at a different viewpoint, even for simple affine transformations such as translation). However, this ability could be induced by pre-training the model on the specific transformation of interest, and this would transfer to unfamiliar classes. For example, a

network pretrained to classify images in which the objects are randomly rotated, develop on-line invariance to rotation, even for novel objects and novel classes. This partially extends even in transformation of viewpoint, in which the object is rotated in depth.

**Datasets: 2D Affine Transformations and Viewpoint Transformations**. We provide a separate dataset for affine transformations (rotation in picture plane, translation, scale, and shear) and viewpoint invariance (rotation in depth). The configurable parameters allow for a fine-grained analysis of the effect of each transformation. For each transformation dimension, the user can chose one or multiple ranges of training and testing. As in the previous datasets, the user can specify any folder containing line-drawings or silhouettes (or any image with a clear contour on a white background). For the viewpoint invariance dataset, we use the ETH-80 dataset (Chen et al., 2020a)[5], which contains 8 categories (apples, cars, cows, cups, dogs, horses, pears and tomatoes), each consisting of 10 object instances, and each object captured from 41 different viewpoints. For the pre-generated dataset, we avoided including views straight from the top. The configurable parameters allow the user to generated dataset only within a specific azimuth and inclination range.

For both the 2D transformation and viewpoint datasets, there are several ways to test whether a DNNs possess online invariance to transformation. First, a DNNs (not necessarely pretrained) could be trained on un-transformed images (e.g. with the object always in the center, unrotated, unscaled, or from a standard viewpoint). It could then be tested on various transformations of these objects. This could be used to establish whether the network is architecturally invariant to some transformations. Several pre-training steps could be used to test how the training environment affects performance. Another approach avoids training on the target classes (either because we want to test a pretrained network without altering its weights, or because we want to test the network on unfamiliar classes): a similarity judgments analysis is performed on transformed versions of the same object, and is compared to the similarity of different objects. A human-like DNN will have internal activations that are more similar for same objects across transformations, than for different objects. This is the approach used in Biscione & Bowers (2021; 2022).

### C.3.6 Same/Different Task

**Psychological Significance.** Human shape representations not only support object recognition but also a wide variety of additional functions, including visual reasoning. Perhaps the simplest form of visual reasoning is tested in the same/different task – judging whether two shapes are identical apart from their spatial location. Although DNNs can solve the same/different task when training and test images are highly similar to one another, performance drops when training/test images are dissimilar (Puebla & Bowers, 2022). By contrast, humans can make same/different judgements for any visual patterns as long as they are perceptible.

**Same/Different Dataset.** The dataset was extracted from (Puebla & Bowers, 2022). This dataset is composed of 10 conditions. Each image consists of two items placed randomly on the canvas. The two items can be either the same shape or a different shape and cannot overlap. Each condition consists of a different type of item used. By default, the items are composed of white strokes with no fill on a black background. See Figure 14 for a summary of all the conditions.

The suggested testing methodology for this condition is slightly different than all other methods, and consists of training a DNN (not necessarily pre-trained) on a subset of conditions, and testing it on a different subbset (as in Puebla & Bowers, 2022).

## D  Supplementary Materials: Accessibility and Licensing

Pre-generated datasets are accessible for viewing and download on Kaggle in both a comprehensive version and a "lite" version, which can be found at large and "lite" respectively. `Croissant` metadata for both versions are automatically generated by Kaggle and can be

---

[5]`https://github.com/chenchkx/ETH-80/tree/master`

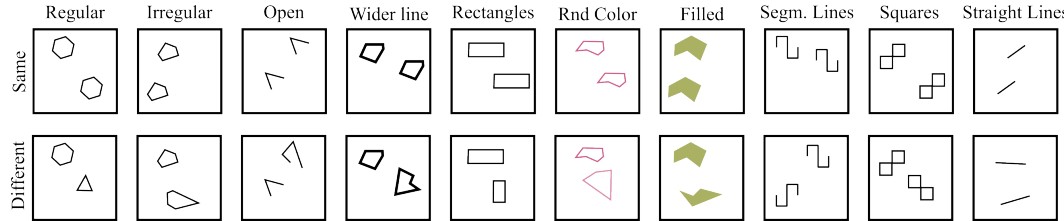

Figure 14: Illustration of the ten conditions used for the Same/Different task. Two items can be either the same or different shapes up to translation. For the 'straight lines' condition, the "same/different" dimension considered is the line orientation (with length kept fixed).

downloaded from the Kaggle page. These datasets comprise a collection of folders containing `.png` images and associated metadata. Notably, each dataset's root folder includes a `.csv` file containing metadata for each individual image, as well as a `.toml` file detailing the configuration parameters used in dataset generation. This structure enables precise replication of each dataset published on Kaggle using our provided codebase. The dataset is made available under the CC 1.0 Universal license.

The codebase for regenerating the dataset with varying parameters is hosted on GitHub, licensed under MIT. Comprehensive documentation on dataset generation procedures is included in the main README file within the repository. Additionally, we provide code for evaluating the datasets using three distinct testing methods: out-of-distribution classification, similarity analysis, and decoder approach. Each method is documented in a dedicated README file within the corresponding folder, such as `src/utils/similarity_judgment/REAMDE.md`, and examples are provided. The appendix in the main text offers detailed descriptions of each dataset, emphasizing that each dataset can be regenerated by specifying a set of parameters. The meaning of each parameter for each dataset is accessible via the `-h` flag and is also provided in the accompanying `.toml` file used for dataset generation. The entire dataset generation process is tested on Windows Server 2022 (Windows 10/11), macOS 14.4.1, and Ubuntu 20.04. Authors assume full responsibility in the event of any rights violations.

