# OpenReview forum: "MindSet: Vision. A toolbox for testing DNNs on key psychological experiments"
_ICLR.cc/2025/Conference — Submitted to ICLR 2025_

### Official Review · Reviewer_VwBg · 2024-10-25

**Soundness:** 3
**Presentation:** 4
**Contribution:** 3
**Rating:** 8
**Confidence:** 4

**Summary:**

This paper introduces a toolbox for creating 30 static visual tasks to test specific hypotheses about the capabilities of DNNs in comparison to the human visual system, for example how a DNN performs when subjected to crowding. Like the well-established Brain-Score benchmark, this toolbox provides a collection of visual tasks on which any DNN could be evaluated. Unlike Brain-Score, the authors propose to not simply evaluate models on observational data and aggregate performance to obtain a ranking, but to explicitly test specific hypotheses about the abilities of the models, which should be simplified by the proposed toolbox.

**Strengths:**

The paper is well-written and technically sound, as is the code. The proposed tasks are all sensible (and could be extended to include new tasks, should the need arise) and reflect the field well, so the toolbox could be of great practical utility. Because the toolbox makes evaluating models on these different hypothesis tests easy, there is a good chance that it will indeed be used to evaluate new models, which would be beneficial for the community. The core point that the authors want to raise (about testing specific hypotheses, instead of predicting observational data) is also valid and meaningfully adds to the discourse about human-DNN alignment.

**Weaknesses:**

The main weakness of the paper is the lack of experiments and results. The authors only present the toolbox and superficially evaluate a ResNet-152, but they do not derive any new findings or concretely outline how this could be done, which would greatly increase the perceived utility of the toolbox. Another issue is the lack of human reference data: While it is generally known that humans are e.g. sensitive to crowding, the toolbox is missing human reference performance for the default set of stimuli that the authors provide. However, such data could over time be collected and contributed by the community.

**Questions:**

The work presented here is done well, and the decision to accept or reject ultimately comes down to whether one believes that the implementation of this toolbox is a sufficient contribution.

I propose to accept the paper, because I can imagine the practical value in having such a standardized toolbox. It will at least save the community some work, and if standardized procedures for generating these stimuli emerge, results could become more directly comparable. Furthermore, the authors argue for an alternative paradigm in how to evaluate human-machine alignment, which I deem a valuable intellectual contribution, even though this point remains a bit vague. Elaborating more on what this would entail could improve the paper. Overall, the paper does feel a bit thin, lacking both evaluations of a relevant suite of models and human reference data. While the latter might be too much to ask, evaluating a few models (e.g. models scoring high on Brain-Score or human-vs-machine) would have been interesting and justified the submission to ICLR better.

As a general point of discussion: In principle, these tasks could be integrated into Brain-Score, if there was human data to compare models to. While I understand that the authors are (for fair reasons) skeptical of the Brain-Score methodology of averaging performance, it would still be valuable to at least reflect this dimension of human-DNN-(mis)alignment on Brain-Score as well, even if it just highlights how none of the models that do well on predicting neural data show human-like performance on these tasks.


### Nitpicks
* in line 49, there seems to be a broken citation
* in line 108, there should be a space: "to understand"
* in the README of the repository, the link to ImageNet classification is broken (because it should be imagenet_ood_classification)

---

> ### Author Response · Authors · 2024-11-19
> **Review response**
>
> We thank the reviewer for his/her positive comments and score.
>
> **The Reviewer writes:**
> *In principle, these tasks could be integrated into Brain-Score, if there was human data to compare models to.*
>
> We agree, and as noted above, 3 experiments included in our toolkit have already been added to Brain-Score.  We hope that publishing our toolkit will result in more of these experiments being added.
>
> We have fixed the *Nickpicks* that the reviewer has identified. Thank you for pointing out these errors.

---

> > ### Comment · Reviewer_VwBg · 2024-11-27
> >
> > Dear Authors, \
> > I don't have much to add to the lively discussion, because as I suspected, the paper decision hinges on the question of whether the toolbox itself is a sufficiently large contribution to ICLR — a point which I am not entirely sure of myself. I do maintain that it is well-executed, deserves visibility and will probably have an impact by standardising stimuli and making results comparable, hence my initial score. I can imagine that this work will see use and receive quite a few citations, and it feels weird to me to reject a paper that I both think is good work and will be used. But looking at the ICLR guidelines, they do quite clearly focus on novelty of the insights, and I must admit that the paper itself does not provide those (even though future works building on this paper could, and probably will) so maybe ICLR is indeed not the best venue.

---

### Official Review · Reviewer_egun · 2024-11-01

**Soundness:** 3
**Presentation:** 3
**Contribution:** 2
**Rating:** 3
**Confidence:** 4

**Summary:**

This work provides Python code to generate controllable visual stimulus sets from 30 well-known cognitive psychology experiments. The underlying motivation is to provide a hypothesis-driven approach for comparing neural networks to human vision, as an alternative to benchmarking with natural image datasets and associated behavioral or neural responses (e.g., BrainScore). The work also introduces three alternative methods for evaluating model predictions, accompanied by example Python implementations.

**Strengths:**

1. Bridging cognitive psychology and neural network research is a laudable goal. Curating experimental results and making them accessible to the neural network research community is an effective way to achieve this and clearly establishes the motivation for this contribution.

2. The collection of stimulus sets is extensive and diverse.

3. Based on my review, the codebase appears well-designed, well-organized, and well-documented. The paper itself is also clearly written.

**Weaknesses:**

1) The paper describes "datasets"; however, it contains no actual, complete data—only stimuli and stimulus generation scripts. In behavioral research, data typically includes, at a minimum, temporally ordered pairs of stimuli and empirical human responses. Generating stimuli is only the first step.

   A major hurdle in comparing human behavior with neural network models is obtaining high-quality human behavioral data collected in controlled settings. This hurdle is particularly challenging for researchers from engineering departments, where institutional infrastructure and practical know-how for conducting in-person human experiments are often lacking. Since data sharing is relatively new in experimental psychology, many classical results are available only as aggregate data (e.g., bar charts or ANOVA outcomes) rather than as recordings of human responses at the single-trial, single-participant level. Such raw responses may be essential for testing new hypotheses and evaluating statistical significance and reliability.

   Currently, the lack of associated behavioral data severely limits the potential impact of this contribution. For comparison, consider the works by Robert Geirhos et al. on robustness to transformations. These works have been impactful not only for their research questions but also for providing large-scale human response data as a community resource.

2. The theoretical contextualization of this work is relatively narrow and one-sided. The authors motivate their approach by referencing the 2022 BBS target paper by Bowers et al., reiterating some of its arguments (e.g., "...competing on observational datasets that do not
support any conclusions regarding the mechanistic similarity of DNNs and brains"). However, the current paper does not substantially engage with any of the counter-arguments presented in the 29 responses to the Bowers et al. paper. Scientific debate is essential, but it can only be productive when effort is made to consider opposing views.

**Questions:**

1. In my opinion, for this work to be complete and ready for publication, each task should be accompanied by human data, either collected by the authors or curated from open data repositories. This data should be provided in both raw format and aggregated by condition. If the effects are strong and unequivocal, obtaining the relevant data should be straightforward. If the interpretation of the experimental outcomes is more complicated, the availability of detailed behavioral data becomes even more important. I believe that such a substantial revision would make this work highly impactful for the field.

2. A complete description of each experiment should also include experiment-specific instructions for both experimenters and participants, as well as the expected results according to alternative hypotheses. An experiment in psychology is more than just a set of stimuli.

3. Will training a neural network directly to imitate human performance in these tasks produce a model that the authors consider valid? If not, it may be worthwhile to specify the appropriate training distribution for each task.

4. In this context, it might be worthwhile for the authors to discuss the recent paper by Marcel Binz et al., "Centaur: A Foundation Model of Human Cognition," which appears to take a direct data-fitting approach.

Minor: There are multiple issues in the bibliography, including duplicated entries and papers already published in conferences cited as arXiv preprints.

---

> ### Author Response · Authors · 2024-11-19
> **Review response**
>
> **The Reviewer writes:**
> *Will training a neural network directly to imitate human performance in these tasks produce a model that the authors consider valid? If not, it may be worthwhile to specify the appropriate training distribution for each task.*
>
> *In this context, it might be worthwhile for the authors to discuss the recent paper by Marcel Binz et al., "Centaur: A Foundation Model of Human Cognition," which appears to take a direct data-fitting approach.*
>
> In our view, training a model on the data itself is not interesting, but that is what the authors did in the Centaur paper.  Our toolkit is somewhat agnostic on how researchers carry out their research (see general responses).
>
> **The Reviewer writes:**
> *The theoretical contextualization of this work is relatively narrow and one-sided…the current paper does not substantially engage with any of the counter-arguments presented in the 29 responses to the Bowers et al. paper...*
>
> We are not challenging DNN-brain alignment, we are just introducing a toolkit to encourage researchers to explore key findings in psychology. We do not believe introducing this toolkit is theoretically biased, and indeed, the field will still be dominated by the many benchmarks composed of naturalistic stimuli which have not been manipulated to test specific hypotheses.

---

> > ### Comment · Reviewer_egun · 2024-11-22
> >
> > I thank the authors for their response. As the replication crisis has clearly demonstrated, "known results" do not always accurately capture the underlying phenomenology. Furthermore, even for well-established effects, the fine-grained patterns of how stimulus parameters modulate responses are generally unavailable.
> >
> > While I appreciate the idea behind the toolbox, I find it incomplete and not yet sufficient to justify publication at ICLR. For instance, generating stimuli to elicit the Ebbinghaus illusion is trivial. With current programming tools (i.e., LLMs), this task can be completed in less than 10 minutes. In contrast, acquiring actual human size judgments for the different conditions is far less straightforward. To be of real value to the modeling community, the stimuli and stimulus-generation scripts must be accompanied by rigorously collected behavioral data.
> >
> > I therefore maintain my score.

---

> > > ### Author Response · Authors · 2024-11-23
> > > **Further response**
> > >
> > > This response is both disagreeable and misguided. It may well take 10 minutes to generate 1000s of Ebbinghaus stimuli with LLMs, but building this toolkit was a substantial project. If the reviewer worked with our toolkit we expect this would be apparent to him or her. The reviewer is also missing our key contribution, namely, we have used our expertise in psychology to select and explain important experiments that have largely been ignored by the ML community. Although LLMs make it easier to generate stimuli, most ML researchers do not know what stimuli to generate, or the theoretical significance of key findings from psychology.  Currently the field focuses on working with naturalistic images using correlational studies, and our goal is to encourage more research using stimuli that have been manipulated to test specific hypotheses.
> > >
> > > With regards to the replication crisis, we have only included experiments in our toolkit that have been replicated many times. Indeed, most of the experimental effects the reviewer can just see for him or herself by simply looking at the stimuli (all the illusions, the Gestalt effects, line drawings, Weber Law, visual transformation, 3D encoding of line drawings, etc.), and many of the findings are taught in Psychology 101.
> > >
> > > There is also confusion in the reviewer’s reasoning regarding replication.  If the reviewer does not trust the results reported in the psychological literature, why should he or she trust the results of experiments that the reviewer wants us to run?  Indeed, why would it be a good idea to provide a single benchmark dataset for each experiment that many modellers can try to predict if replication is such a problem?  Far better would be the approach we are advocating: Providing the tools for researchers to carry out bespoke experiments that target hypotheses.  And with the scripts we provide, images can be varied in parametric ways to test more specific hypotheses, something that cannot be done if the field focuses on accounting for the most variance on a small set of benchmarks datasets carried out by one group. With different labs collecting datasets on important visual phenomena, the replication (or not) of the findings would be clear, and a wider range of hypotheses can be addressed.
> > >
> > > Indeed, there is a potentially serious replication crisis with the current benchmark approach in NeuroAI.  For example, in the Brain-Score benchmark, we have 100s of models competing to account for the neural recordings in V4 and IT taken from two macaques.  Perhaps there is something idiosyncratic with this dataset, and models will do terribly on data collected from a 3rd macaque presented with the same images, or slightly different images.  Or consider another example. When Xu and Vaziri-Pashkam (2021) ran RSA studies on a better dataset than used by Kriegeskorte, the authors did not replicate key findings, including the hierarchical correspondence between ANNs and visual cortex.  But again, this one dataset has been used over and over.
> > >
> > > We find it extraordinary that Reviewer egun questions our approach because some of the most basic findings from the psychology of vision literature may not be replicable.  Similarly, we find it extraordinary that we need to convince Reviewer wknk that psychological data are relevant to assessing ANN-brain alignment, and further, who thinks that we need to show that some ANNs can account for the experiments in our toolkit (that is, it not enough to report failures of alignment to publish), and concludes “I'd prefer empirical science and/or theory over philosophy for an ICLR paper”.
> > >
> > > Xu, Y., & Vaziri-Pashkam, M. (2021). Limits to visual representational correspondence between convolutional neural networks and the human brain. Nature communications, 12(1), 2065.

---

> > > > ### Comment · Reviewer_wknk · 2024-11-23
> > > > **Response**
> > > >
> > > > I simply think the authors should prove it. If the authors believe that DNNs struggle on these psychological studies, then we should see it after the authors test TIMM models on them. Everything else is conjecture and philosophizing about why biological brains and DNNs are different. Just run the experiments.

---

> > > > > ### Author Response · Authors · 2024-11-23
> > > > > **Final response to wknk**
> > > > >
> > > > > As we have pointed out before there is already lots of evidence that ANNs fail on the experiments in our benchmark. Let us repeat a comment we sent before.
> > > > >
> > > > > “… three experiments included in our toolkit have been added to the Brain-Score benchmark with 21 models already tested. Interestingly, the top-performing models on Brain-Score do poorly on the three experiments”.  Indeed, the two winning entries on the most recent Brain-Score benchmark that judged benchmarks on how *poorly* models used images included in our toolkit.
> > > > >
> > > > > And again, here is a list of experiments that focus on experiments included in toolkit. In most cases models did terribly, and in all cases, models struggled in important ways, with more work needed. Here they are again:
> > > > >
> > > > > Lightness illusions: Mukherjee, A., Paul, A., & Ghosh, K. (2024). Deep learning models for perception of brightness related illusions. Applied Intelligence, 54(21), 10259-10283.
> > > > >
> > > > > The role of parts and relations in object identification: Malhotra, G., Dujmović, M., Hummel, J., & Bowers, J. S. (2023). Human shape representations are not an emergent property of learning to classify objects. Journal of Experimental Psychology: General.
> > > > >
> > > > > Perceptual organisation (Gestalt rules): Biscione, V., & Bowers, J. S. (2023). Mixed evidence for gestalt grouping in deep neural networks. Computational Brain & Behavior, 6(3), 438-456.
> > > > >
> > > > > Müller-Lyer Illusion: Zhang, H., Yoshida, S., & Li, Z. (2023, October). Decoding Illusion Perception: A Comparative Analysis of Deep Neural Networks in the Müller-Lyer Illusion. In 2023 IEEE International Conference on Systems, Man, and Cybernetics (SMC) (pp. 1898-1903). IEEE.
> > > > >
> > > > > Thatcher illusion: Dobs, K., Martinez, J., Kell, A. J., & Kanwisher, N. (2022). Brain-like functional specialization emerges spontaneously in deep neural networks. Science advances, 8(11), eabl8913.
> > > > >
> > > > > Line drawings and silhouettes: Baker, N., Lu, H., Erlikhman, G., & Kellman, P. J. (2018). Deep convolutional networks do not classify based on global object shape. PLoS computational biology, 14(12), e1006613.
> > > > >
> > > > > Same/Different task and visual reasoning: Puebla, G., & Bowers, J. S. (2022). Can deep convolutional neural networks support relational reasoning in the same-different task?. Journal of Vision, 22(10), 11-11.
> > > > >
> > > > > Orientation invariance: Biscione, V., & Bowers, J. S. (2022). Learning online visual invariances for novel objects via supervised and self-supervised training. Neural Networks, 150, 222-236.
> > > > >
> > > > > Global shape using Frankenstein images: Baker, N., & Elder, J. H. (2022). Deep learning models fail to capture the configural nature of human shape perception. Iscience, 25(9).

---

> > > > > > ### Comment · Area_Chair_oGoF · 2024-11-25
> > > > > >
> > > > > > Dear Reviewer,
> > > > > >
> > > > > > The authors have provided their responses. Could you please review them and share your feedback?
> > > > > >
> > > > > > Thank you!

---

> > > > ### Comment · Reviewer_egun · 2024-11-25
> > > >
> > > > The authors’ response seems to misconstrue my point about the replication crisis. The finding that roughly 50% of cognitive psychology experiments from the Open Science Collaboration 2015 study failed to replicate does not diminish the importance of behavioral data. Instead, it highlights the need for additional data. This need stems from "results" often being drawn from underpowered experiments and opaque analytical procedures, compounded by insufficient or nonexistent data sharing and publication bias.
> > > >
> > > > Although I agree with the authors that many of the experiments described in their paper are century-old classics and can be considered well-replicated, some included studies are more recent. For well-replicated studies, curating empirical behavioral results to complement the stimuli with actual data should be straightforward. For newer studies, acquiring such data is even more important. All included studies would benefit from quantitative, fine-grained data, as qualitative descriptions of effects often lack the detail required by modelers.
> > > >
> > > > Given these gaps, offering stimuli or stimulus-generation scripts without associated data seems unlikely to advance neuroAI research. Naturally, it would be ideal if everyone in the field could rigorously collect behavioral data; however, as the authors know, there are considerable barriers to this.
> > > >
> > > > The critiques of Brain-Score raised by the authors are irrelevant to the evaluation of this manuscript. The key question is whether this submission meets the significance threshold for ICLR. At present, this contribution provides a well-written review of foundational psychological findings, some imbalanced commentary about the neuroAI field, and 30 scripts for parametrically generating experimental stimuli required for replicating these findings. In my opinion, this does not meet the significance threshold for ICLR.
> > > >
> > > > To clarify, I am sympathetic to the general direction of this work; however, it remains incomplete. If the authors wish to provide a theory-driven alternative to Brain-Score, it must incorporate actual behavioral data.

---

### Official Review · Reviewer_yu8u · 2024-11-04

**Soundness:** 3
**Presentation:** 4
**Contribution:** 4
**Rating:** 3
**Confidence:** 5

**Summary:**

This is a very interesting and well timed paper that introduced a dataset full of interesting psychological visual phenomena to test and compare humans and machines in a more precise and well-rounded manner than just accuracy/performance on a classification dataset (which is classical in computer vision & machine learning). The paper is extremely well written and articulated the need for a dataset like this one, and the authors do a spectacular job in citing all the relevant literature in this new type of "NeuroAI" wave where humans and DNN's are being compared. The only main weakness I see is that I wish there were more evaluations, or at least one solid one which is why I have reserved my vote for a marginally above threshold. Nevertheless, I do think that all in all this paper should be accepted provided a more extensive evaluation of at least one Neural Network with more detail -- that I did not see in the Appendix (although the Appendix does do a great job in covering all the vision science phenomena and illusions that are to be studied/tested by humans and machines).

----------

**Upgrade*

I have downgraded by score to 3 after reading all the reviews from different reviewers and I think the consesus holds that even though the papers intention is very insightful and unique, the lack of experiments and empirical validation for human-machine visual alignment undersells the true potential value of this paper. I think adding those basic experiments (only even 1 to 3 architectures) can put it at a high Oral-worthy position for a future CV/ML conference.

**Strengths:**

* The paper is extremely well written. Despite having a background at the intersection of human and machine vision as a reviewer and being quite familiar with nearly all the works cited, the authors have done a great job in articulating why this dataset is important **for machine vision & computer vision** hence the proper fit for ICLR.
* The paper covers a lot of great literature at the intersection of machine vision and human vision, in addition to papers that try to link both phenomena. I think this paper could serve as a great "reference" to read for many early graduate students who would like to learn more about the topic that the authors are exploring and justifying its importance.
* The rigor in which the paper all 30 psychological findings in the main body and appendix is superb.

**Weaknesses:**

The greatest weakness I see in this paper is that there is little to no evaluations. There is barely only two figures to evaluate the ResNet-152, and the paper leaves me with a bittersweet feeling of wanting to know how well did the ResNet-152 do compared to the average human on the collection of all the images in the toolbox.

**Questions:**

Is there anyway we can get a figure of how the ResNet-152 does for all 30 psychological findings, as if we could get a row of 1 model entry as if this was Brain-Score? I think this would really drive the paper home and come full circle so that readers from computer vision and machine learning who do not know anything about neuroscience (or assume that machines see just like humans) -- can be surprised and find this paper even more valuable.

**Details Of Ethics Concerns:**

None.

---

> ### Author Response · Authors · 2024-11-19
> **Review response**
>
> Again, we very much appreciate the positive comments and score of the reviewer.
>
> **The Reviewer writes:** *Is there anyway we can get a figure of how the ResNet-152 does for all 30 psychological findings, as if we could get a row of 1 model entry as if this was Brain-Score?*
>
> The problem with doing this is that it would give the impression that we are trying to assess ResNet-152 as a model of human vision, which we are not.  Instead, we only used ResNet-152 to illustrate how our methods of evaluation (e.g., Euclidian similarity and decoder methods) can be applied to our experiments. And if we did test ResNet-152 on all experiments, we expect another reviewer might say: “have you tried models X, Y, and Z”? As noted above, ResNet-152 does poorly on 5 of the experiments included in our Toolkit (2 from our paper and 3 reported in Brain-Score), and no model currently accounts for many of the results of the experiments included in our toolkit (for example, Jacob et al., 2021).
>
> Jacob, G., Pramod, R. T., Katti, H., & Arun, S. P. (2021). Qualitative similarities and differences in visual object representations between brains and deep networks. Nature communications, 12(1), 1872.

---

> > ### Comment · Reviewer_yu8u · 2024-11-26
> > **Follow-Up Impression from General Response and Other Reviews**
> >
> > Dear Authors,
> >
> > Thank you for your reply and the great synthesis of feedback. I am siding more with reviewer wknk who has also highlighted and seen the value of this paper but is voting reject due to lack of any experiments (even reviewer VwBg voting with 8 has this bitter-sweet feeling too). My sentiment is similar, without a single entry and proper evaluation of any single neural network, then it leaves me frustrated to empirically say that a classical Model does perform poorly on such images. I think the condition for acceptance without making it over exhaustive would have been to run 3 classical models such as a fully connected network, a CNN (like a ResNet), and a Transformer model, and empirically showcase that these 3 canonical architectures all present shortcomings regarding human-machine visual alignment and re-emphasize that the point is not to create a model that is better than others, but to showcase the limitations -- such as several of the works for example done by Geirhos, Feather, Deza, Deny or Conwell.
> >
> > I will unfortunately have to downgrade my score to 3. A follow-up submission of this paper even with an additional small baseline would put this paper at a superb position likely for an Oral at a future ML conference (ICML/CVPR/ICLR/NeurIPS), and I think that would be more beneficial for the community to help support the claim that even classical standard vision models do poorly with human perceptual alignment.

---

> > > ### Author Response · Authors · 2024-11-28
> > >
> > > We appreciate the positive comments, but do not understand why you have concluded:  “…without a single entry and proper evaluation of any single neural network, then it leaves me frustrated to empirically say that a classical Model does perform poorly on such images”.  As noted in our response to reviewer wknk, three benchmarks based on images in our toolkit have recently been added to Brain-Score, and 21 models all did poorly (2 of the benchmarks using stimuli from our toolkit won the competition).  In addition, we have provided references to 9 papers that systematically explored different image conditions included in our benchmarks, and again, all models fail in important ways on these image datasets. And Bowers et al. (2023) in Behavioural and Brain Sciences systematically summarized how ANN account for few findings in psychology.  It has already been demonstrated that current ANNs do poorly on classical psychology experiments, and if we test three standard ANNs on all 30 experiments they will fail on many of them.  We will then be open to the criticism that some other models would succeed and that we have only provided a superficial falsification of a small set of models. And that would be a valid criticism.
> > >
> > > Our goal is to change how ANN-brain alignment is assessed.  Rather than compete on accounting for the most variance in an observational dataset, we want to encourage researchers to:  1) Focus on psychological experiments that manipulate independent variables, and 2) carry out systematic studies of these phenomena.  We want to get away from this practice of having a standardized behavioural benchmark that researchers compete on and instead carry out systematic investigations of important psychological phenomena, in line with the 9 articles we include in our response to reviewer wknk.
> > >
> > > The NeuroAI field is so focused on assessing models based on benchmarks (most often composed of observational datasets), and alternative approaches that do not follow this format are hard to publish in the ML literature.  This is a serious problem in our view, and we briefly discuss this further in our general comments to the editors above.

---

### Official Review · Reviewer_poc1 · 2024-11-04

**Soundness:** 4
**Presentation:** 3
**Contribution:** 3
**Rating:** 8
**Confidence:** 4

**Summary:**

The paper introduces MindSet a toolbox for testing DNNs on more than 30 psychological findings.  Including  test for low-level vision,  and high level vision such as visual illusions  and  shape and object recognition. This toolbox, will help authors test on a controlled environment how brain-like are these.

**Strengths:**

I really appreciate the effort that went into creating this toolbox. I think it could greatly benefit the community, especially since many recent tests have focused increasingly on performance in neural alignment, often overlooking functional alignment beyond accuracy alone. The presentation is well-executed, and I particularly enjoyed the failure modes discussed in the supplementary material. My only suggestion would be to consider moving some of these examples into the main text to further highlight the value of this toolbox.

**Weaknesses:**

Perhaps my only question is that it seems that is largely synthetic images on black and white,  which makes it more controllable, but not sure if this would also constrain the kind of models and training routines that can be used in the test, because most models would be trained on complicated backgrounds, such as naturalistic scenes.

**Questions:**

* Would the types of images in the benchmark constrain the kinds of models or training data that people can use, given that most models are usually trained on more crowded scenarios?

---

> ### Author Response · Authors · 2024-11-19
> **Review response**
>
> We very much appreciate the positive comments and score of the reviewer.
>
> **The Reviewer writes:** *Would the types of images in the benchmark constrain the kinds of models or training data that people can use, given that most models are usually trained on more crowded scenarios?*
>
> That is a good question for future research.  But if models only succeed when trained on modified data, that would be a problem.  For instance, children can classify line drawings without any training on line-drawings.  If DNNs only succeed when trained on line-drawings, this is a problem when claiming that DNNs are like humans.

---

### Official Review · Reviewer_wknk · 2024-11-04

**Soundness:** 1
**Presentation:** 2
**Contribution:** 2
**Rating:** 3
**Confidence:** 4

**Summary:**

They authors introduce MindSet, a benchmark of image datasets and experiments to compare visual perception of humans and models. The benchmark spans multiple levels of perception, from low and mid-level vision to object recognition and visual reasoning. What differentiates this benchmark from others is that it combines experimental manipulations with each image dataset. For example, in a version of a same/different task that's been studied over the years, the authors test the task as they change object features, such as the color, open/closed-ness of the shapes, etc. The benchmark is comprehensive, albeit scattershot, and the authors also discuss a variety of ways to evaluate models on their benchmark.

**Update**

After discussing with the authors, I've decided to keep my score. I think they have great points about how NeuroAI should shift incorporate a broader set of psychological experiments as well as perturbations to build the next generation of models of biological vision. I also think that the experiments and datasets described in this paper could be one part of that story. But I disagree with the authors that the current paper constitutes a sufficient work package for ICLR. What I would have liked to see are benchmarks on the TIMM library (https://github.com/huggingface/pytorch-image-models) to show empirically how today's models, designed by computer scientists, respond on these experiments. The authors point out that some of that work has been done in the past, but there is not central benchmark for these models, and the authors must back up their claims of misalignment and failure of today's models with evidence. I believe the bar for publication in a journal/conference that is not a review should be experiments like these, even if they are null results. If those experiments were here I would recommend acceptance. As they are not, I do not think this paper should go through and will keep my score where it is.

**Strengths:**

I really like the emphasis on perturbation, which is an essential missing ingredient in popular benchmarks like BrainScore. This indeed appears to be the main — and IMO a powerful — motivation for the paper. To push computational vision and neuroscience into a new and more effective regime for modeling brain data.

The benchmark itself is comprised of many different interesting experiments taken from psychology and neuroscience. I appreciate the goal of searching for a single model that can explain everything from lightness illusions to visual reasoning.

**Weaknesses:**

The biggest problem with this paper is that it does not make a clear contribution. None of the experiments are novel. There's no actual benchmarking done on models. It's a scattershot approach which is probably the only way this could be done, but there's no control across the different benchmarks — i.e., a single set of visual primitives used to test low-and mid-level vision and visual illusions vs. another set for shape/object recognition. I don't know what to take from this paper other than it's reasonably argued that the current state-of-the-art is insufficient and perturbations would be a better way of advancing models of human vision. Not much more to say than this. What the paper needs is the following:

- A large scale survey of existing models. How well do they capture human data for each experiment?
- If there's no positive result in existing models, provide a model that has a positive result.
- Ideally, and as mentioned above, data is held constant while you change task. For example, could you test relational vs. coordinate change, NAP vs. MP, Muller-Lyer illusion using the same constituent visual components? That is, the number of "on" pixels is the same in each example, but the perceptual effect those pixels have is fundamentally different? This would at least make it possible to plot the experimental results of each of these experiments against each other. This is more of a thought than a must have, given the wide variety of experiments, but IMO it is an essential part of testing models in order to ensure that task rather than statistics are driving a given result.

Another problem I see is in the similarity judgement analysis summarized in Fig 2. The authors use euclidean distance between stimuli to compare their representations at different layers of a network. Why use this instead of the decoder method? There's no reason why euclidean distance is the "right" metric for making decisions at a given layer. Moreover, I don't believe that the bar-and-whisker plots back up the authors' assertion of a difference in same/different shape conditions in early vs. late layers.

**Questions:**

See weaknesses.

---

> ### Author Response · Authors · 2024-11-19
> **Review response**
>
> **The reviewer writes:**
> *The biggest problem with this paper is that it does not make a clear contribution. None of the experiments are novel.*
>
> Our toolkit is designed to encourage researchers to build models that capture key psychological findings. While the experiments and phenomena are not novel, a dataset covering these important effects and scripts to easily generate more samples ***is***. Not only is it novel for researchers testing DNNs, to whom many of these findings will be novel, it is also novel for psychologists as this type of dataset has never been collected in a similar manner before.
>
> **The reviewer writes:** *Ideally, and as mentioned above, data is held constant while you change task. For example, could you test relational vs. coordinate change, NAP vs. MP, Muller-Lyer illusion using the same constituent visual components?*
>
> We do not understand this point.  The norm in science is to have different experiments to test different hypotheses, and there is no requirement to use one set of stimuli to test different phenomena.  We also do not think it would be possible to keep the data constant to test different phenomena.
>
> **The reviewer writes:**
> *Another problem I see is in the similarity judgement analysis summarized in Fig 2. The authors use euclidean distance between stimuli to compare their representations at different layers of a network. Why use this instead of the decoder method? There's no reason why euclidean distance is the "right" metric for making decisions at a given layer. Moreover, I don't believe that the bar-and-whisker plots back up the authors' assertion of a difference in same/different shape conditions in early vs. late layers.*
>
> As we note in the paper, there is no single *correct* way to test a model, and there may be better methods that we have not yet considered.  Both the similarity analyses and decoder methods have been used in the literature, each has its own advantages, and our goal was simply to illustrate how both methods can be used on stimuli from our toolkit.  We don’t understand the reviewer’s objection to our characterization of our findings, we are simply reporting what we found descriptively (does the reviewer want us to provide a statistical interaction?; we can provide this in a revision).  But again, we are including the experiment as an illustration as to how you can assess model performance using similarity analyses, we are not providing a thorough test of ResNet152 or DNNs in general.
>
> **The Reviewer writes:**
> *If there's no positive result in existing models, provide a model that has a positive result.*
>
> The view that falsification is not enough and that *positive evidence* (that is, evidence that DNNs are like the brain) is needed to publish is common in the field, for example, see Bowers et al. (2023).  We disagree regarding the importance of falsification, but in any case, we are not trying to falsify or even evaluate models in this manuscript. We are instead providing a set of criteria that assess whether a model is on the right track as a model of human vision, along with an accompanying toolkit to allow researchers to evaluate a model's performance on those criteria.
>
> Bowers, J. S., Malhotra, G., Adolfi, F., Dujmović, M., Montero, M. L., Biscione, V., ... & Heaton, R. F. (2023). On the importance of severely testing deep learning models of cognition. Cognitive Systems Research, 82, 101158.

---

> > ### Comment · Reviewer_wknk · 2024-11-19
> > **Response**
> >
> > **Our toolkit is designed to encourage researchers to build models that capture key psychological findings. While the experiments and phenomena are not novel, a dataset covering these important effects and scripts to easily generate more samples is. Not only is it novel for researchers testing DNNs, to whom many of these findings will be novel, it is also novel for psychologists as this type of dataset has never been collected in a similar manner before.**
> >
> > It's a challenging position for authors to be in to have to argue that their data is important without actually testing it or showing that it is. Right now my evaluation of the paper is purely based on my subjective opinion that I don't think combining all these datasets together in one package is a significant contribution. What I'd like to see is the authors force me to make an objective opinion about experimental design and results — if we were operating in that regime, and if the experiments were run properly, this would be a no-brainer accept. As is, I don't see it the contribution.
> >
> > **We do not understand this point. The norm in science is to have different experiments to test different hypotheses, and there is no requirement to use one set of stimuli to test different phenomena. We also do not think it would be possible to keep the data constant to test different phenomena.**
> >
> > I agree I am out over my skis a bit with this comment. What I wanted to see were modeling results. The most controlled way to understand a model's success/failure at capturing some phenomena is to hold constant data while varying task. There's no modeling experiments here so I agree my comment doesn't need to be addressed. It would be great to have those controlled results in the future, though.
> >
> > **As we note in the paper, there is no single correct way to test a model, and there may be better methods that we have not yet considered. Both the similarity analyses and decoder methods have been used in the literature, each has its own advantages, and our goal was simply to illustrate how both methods can be used on stimuli from our toolkit. We don’t understand the reviewer’s objection to our characterization of our findings, we are simply reporting what we found descriptively (does the reviewer want us to provide a statistical interaction?; we can provide this in a revision). But again, we are including the experiment as an illustration as to how you can assess model performance using similarity analyses, we are not providing a thorough test of ResNet152 or DNNs in general.**
> >
> > Aggregating existing experimental datasets + metrics strikes me more as a review than an original contribution. My hope is that there would at least be some kind of direction on what the "right" approach to evaluation is. I see that's asking for a different paper than the authors have written.
> >
> > **The view that falsification is not enough and that positive evidence (that is, evidence that DNNs are like the brain) is needed to publish is common in the field, for example, see Bowers et al. (2023). We disagree regarding the importance of falsification, but in any case, we are not trying to falsify or even evaluate models in this manuscript. We are instead providing a set of criteria that assess whether a model is on the right track as a model of human vision, along with an accompanying toolkit to allow researchers to evaluate a model's performance on those criteria.**
> >
> > I'd prefer empirical science and/or theory over philosophy for an ICLR paper. I think BBS is a great spot for this kind of paper.

---

> > > ### Author Response · Authors · 2024-11-21
> > > **Further response**
> > >
> > > We agree with the reviewer that not all datasets are useful for assessing ANN-brain alignment and that it is incumbent on us to show that our article and toolkit is a contribution.  In this case, the usefulness of our image datasets has already been established by the long history of research in experimental psychology that provides insights into how human vision works. Any ANN model of human vision needs to account for these findings. There is also a need for more researchers to assess ANN-brain alignment using psychological findings, and our article reviews 30 psychological findings from low- to high-level vision, and the toolkit facilitates that testing of ANNs on these experiments.
> > >
> > > There are at least two general ways our toolkit can facilitate research going forward. First, it can be used to collect benchmark datasets that anyone can use to test and compare models. Indeed, as noted above, three experiments included in our toolkit have been added to the Brain-Score benchmark with 21 models already tested. Interestingly, the top-performing models on Brain-Score do poorly on the three experiments.
> > >
> > > Another approach is to carry out systematic studies on individual experiments, manipulating images as required (as made possible in our toolkit) to provide severe tests of ANNs.  There is a small but growing set of studies adopting this approach (mostly carried out by psychologists thus far), with individual papers focusing on specific phenomena, with some examples listed below.  In either case, we think the toolkit is an important contribution by encouraging more focus on psychological data.
> > >
> > > To give just one example of the value of psychological experiments in assessing ANN-brain alignment, consider the winning entry by Nicholas Baker at the recent Brain-Score competition.  He used the Frankenstein images included in our toolkit designed to assess whether ANNs show similar configural effects (global shape sensitivity) to humans. Not only did the 21 ANNs included in the competition do poorly on these stimuli overall, Baker and Elder (2022) ran a series of simulation studies to try to try and address the misalignment.  This is from their abstract of their paper referenced below:
> > >
> > > *Modifications to training and architecture to make networks more brain-like did not lead to configural processing, and none of the networks were able to accurately predict trial-by-trial human object judgements. We speculate that to match human configural sensitivity, networks must be trained to solve a broader range of object tasks beyond category recognition.*
> > >
> > > That is, future work is needed to address this misalignment, and our toolkit will help in this endeavor.  And note, this limitation was only observed when running experiments that manipulated the configural properties of mages.  The standard approach in Brain-Score that largely focused on naturalistic images could not reveal this limitation.
> > >
> > > Articles that focus on experiments included in toolkit. In all cases more work is needed.
> > >
> > > Lightness illusions: Mukherjee, A., Paul, A., & Ghosh, K. (2024). Deep learning models for perception of brightness related illusions. Applied Intelligence, 54(21), 10259-10283.
> > >
> > > The role of parts and relations in object identification: Malhotra, G., Dujmović, M., Hummel, J., & Bowers, J. S. (2023). Human shape representations are not an emergent property of learning to classify objects. Journal of Experimental Psychology: General.
> > >
> > > Perceptual organisation (Gestalt rules): Biscione, V., & Bowers, J. S. (2023). Mixed evidence for gestalt grouping in deep neural networks. Computational Brain & Behavior, 6(3), 438-456.
> > >
> > > Müller-Lyer Illusion: Zhang, H., Yoshida, S., & Li, Z. (2023, October). Decoding Illusion Perception: A Comparative Analysis of Deep Neural Networks in the Müller-Lyer Illusion. In 2023 IEEE International Conference on Systems, Man, and Cybernetics (SMC) (pp. 1898-1903). IEEE.
> > >
> > > Thatcher illusion: Dobs, K., Martinez, J., Kell, A. J., & Kanwisher, N. (2022). Brain-like functional specialization emerges spontaneously in deep neural networks. Science advances, 8(11), eabl8913.
> > >
> > > Line drawings and silhouettes: Baker, N., Lu, H., Erlikhman, G., & Kellman, P. J. (2018). Deep convolutional networks do not classify based on global object shape. PLoS computational biology, 14(12), e1006613.
> > >
> > > Same/Different task and visual reasoning: Puebla, G., & Bowers, J. S. (2022). Can deep convolutional neural networks support relational reasoning in the same-different task?. Journal of Vision, 22(10), 11-11.
> > >
> > > Orientation invariance: Biscione, V., & Bowers, J. S. (2022). Learning online visual invariances for novel objects via supervised and self-supervised training. Neural Networks, 150, 222-236.
> > >
> > > Global shape using Frankenstein images: Baker, N., & Elder, J. H. (2022). Deep learning models fail to capture the configural nature of human shape perception. Iscience, 25(9).

---

> > > > ### Comment · Area_Chair_oGoF · 2024-11-25
> > > >
> > > > Dear Reviewer,
> > > >
> > > > The authors have provided their responses. Could you please review them and share your feedback?
> > > >
> > > > Thank you!

---

> > > > > ### Comment · Area_Chair_oGoF · 2024-11-30
> > > > >
> > > > > Dear Reviewer,
> > > > >
> > > > > The authors have provided their responses. Could you please review them and share your feedback?
> > > > >
> > > > > Thank you!

---

### Author Response · Authors · 2024-11-19
**General response**

It is striking how divided the responses are to the submission, with two 8s, two 3s, and a 6. We think this reflects the fact that our toolkit stems from a fundamentally different approach to studying DNN-brain alignment, and some basic misunderstandings of that approach.  When scores are distributed like this, an average score provides a misleading overall assessment, and it disadvantages controversial articles that are more likely to receive contrasting responses like this.  The critical question is whether the higher or lower scores better characterize the contribution.

Before responding to the individual reviewers below we would like to (1) highlight how our approach to evaluating DNN-brain alignment in vision is quite different than standard practice and (2) address the two most important criticisms that were common across multiple reviews. In both cases, there are important misunderstandings that are contributing to the lower scores.

A standard practice in NeuroAI for assessing DNN-brain alignment is to develop benchmark composed of a set of naturalistic images paired with associated behavioural or brain responses so modellers can compete in accounting for variance explained on that dataset.  More recently, some benchmarks have included artificial stimuli manipulated in theoretically motivated ways, such as the most recent version of Brain-Score.  But to date, the field is characterized by attempting to build models that account for as much variance on a relatively small set of benchmarks, with relatively little consideration of psychological findings.  By contrast, the MindSet: Vision toolkit includes stimuli from 30 psychological experiments where the qualitative main effects of manipulations are well-known, theoretically significant, and robust.  For instance, humans are reliably subject to a specific visual illusion under certain conditions. Our toolkit can be used to build models that account for a wide range of critical psychological findings at a qualitative level (e.g., does a model show a significant illusion or not). Both approaches have their place, but the latter method is rarely used at present. Our goal is to highlight to the ML and NeuroAI communities a set of critical psychological experiments as well as the tools to facilitate the testing of models on these experiments where the qualitative effects are well established.

---

> ### Author Response · Authors · 2024-11-19
> **General response to reviewers**
>
> With regards to the reviews, one repeated comment is that we should have assessed how multiple existing models perform on the experiments included in the toolkit (for Reviewer wknk this is a critical weakness that led to a low score). However, our goal is to encourage the field to test DNNs using classic experiments from psychology, explore ***how*** and ***why*** models succeed and fail, rather than simply evaluating and optimizing towards one score.
>
> To this end, we have identified and described the significance of 30 psychological experiments, provided image datasets and scripts to generate image datasets in a format that can be easily used to train and test DNNs. We also suggest possible methods of testing whether models exhibit these phenomena (illustrated by testing ResNet152), but we do not proscribe a single way of going about this.
>
> Furthermore, it is not possible to meaningfully and systematically test many models on 30 different experiments in a single paper.  Imagine if we assessed how ResNet152 performed on many or most of the experiments.  If it did poorly on some or most of the conditions, a reasonable response would be that some other model might succeed. We don’t want to provide a trivial falsification of a small set of models. But perhaps it is worth noting that 3 experiments from our toolbox have already been included in the recent [Brain-Score competition](https://www.brain-score.org/competition/) that assessed performance of 21 models. The datasets won both behavioural track prizes in which the goal was to get poor DNN-human alignment. At the same time, when researchers develop DNNs that do succeed on the experiments included in the toolkit, it will suggest that the models are implementing key theoretical processes in human vision (e.g., a model that supports Gestalt principles is organizing elements in scene in a human-like way).  That is, the experiments in our toolkit provide an important and challenging step in evaluating ANN-brain alignment.
>
> The bottom-line: Our goal is not to make theoretical claims regarding models but to make researchers aware of critical experiments in psychology and facilitate testing of models on these experiments going forward.  A good model of human vision needs to account for these key phenomena.
>
> The other repeated comment is that we should have collected behavioural data for our 30 experiments (for Reviewer egun this is a critical weakness that led to a low score). We agree that collecting human data on all the experiments would be a valuable contribution, and as mentioned, three experiments from MindSet have been carried out and included in the Brain-Score benchmark.  But the critical point is that we already know the qualitative results from all 30 experiments, and current DNNs struggle to account for these findings.  Accordingly, the toolkit can already be used to provide severe tests of models without collecting new behavioural data. Furthermore, the phenomena in MindSet change in magnitude predictably depending on how the datasets are generated. The impact of various parameters is well documented and thus allows for testing whether these phenomena change in models as they do in humans based on these parameters. Finally, once DNNs do account for a given experiment at a qualitative level, the toolkit will make it easier for researchers to carry out the relevant experiments if their goal is to assess percentage of variance explained.
>
> The bottom-line: The qualitative results of all the 30 experiments are already known, and current DNNs already struggle to account for these effects.  Accordingly, our toolkit can be used as is to provide severe testing of DNNs.
>
> We have added a new section to our submission entitled “Limitations” to clarify our goals and contributions.
>
> Please find responses to other specific comments in review responses for each reviewer.

---

### Author Response · Authors · 2024-11-28
**General comment to editors summarizing our responses: Part 1**

We just wanted to highlight our more important points and disagreements with the reviewers after a lengthy exchange.  In all the back-and-forth it is easy to lose track of the fundamental issues.  We first consider the most important criticisms of the reviewers, and then describe why we think the toolkit is an important contribution.

The three negative reviewers all see promise in our work, but 1) they all think we need to test a set of ANNs on all the experiments and 2) some reviewers think we need to collect behavioral datasets for some or all of our experiments.  The reviewers make three arguments to support their conclusions.

First, some Reviewers are unconvinced that ANNs will struggle on the psychological experiments included in the toolkit, and without this, it is not clear whether the experiments provide a useful test of models. But it is already well demonstrated that ANNs do poorly on a wide variety of psychological experiments.  For example, three benchmarks based on images in our toolkit have recently been added to Brain-Score, and 21 models all did poorly (2 of the benchmarks using stimuli from our toolkit won the competition).  We have also provided references to 9 articles that systematically explored different image conditions included in our benchmarks, and again, all models fail in important ways on these image datasets. And Bowers et al. (2023a) in Behavioural and Brain Sciences systematically summarized how ANN account for few findings in psychology.

Imagine we had carried out the requested simulation studies and assessed how well a few ANN performed on a single experiment in each of the 30 stimulus conditions.  The ANNs would fail on many of the conditions (see above), and then we would be open to the criticism that some other (untested) models would succeed.  That is, we will only have provided a superficial falsification of a small set of models. And that would be a valid criticism.  Furthermore, reviewer wknk does not think reporting failures of ANNs is adequate, writing: “If there's no positive result in existing models, provide a model that has a positive result”.  This view is commonplace, as NeuroAI has a strong bias against accepting papers that highlight limitations of ANNs (Bowers et al., 2023b).  This is another reason not to provide a superficial falsification of ANNs.  And in any case, our goal is not to falsify ANNs, but to encourage researchers to adopt a different way of assessing ANN-brain alignment to make better models of biological vision.

Second, some of the reviewers think we need to carry out behavioral studies to make it easy for researchers to assess how much variance models account for, similar to the Brain-Score approach.  The assumption is that trial-level data provides a better assessment of a model than the condition-level results reported in psychological studies. But this is mistaken. The entire point of manipulating independent variables is to test specific hypotheses.  By contrast, by assessing amount of variance explained, you do not know whether the model captures the impact of the independent variable.  For instance, a model could account for a substantial amount of variance and not account for the manipulation at all (of course, if the model accounted for 100% of variance, it would necessarily account for the manipulation, but short of that, there are many other sources of uncontrolled variation that a model could account for without accounting for the manipulation).  The Brain-Score approach makes sense when analyzing observational datasets in which no variables have been manipulated, but in our dataset, the key issue is whether ANNs can capture key effects of the manipulation.  We know the impact of the manipulations from past psychological research, and we know that current ANNs struggle with these effects.  Our toolkit can support many research programs attempting to build models that account for these key properties of human vision.

---

> ### Author Response · Authors · 2024-11-28
> **General comment to editors summarizing our responses: Part 2**
>
> Third, reviewer egun thinks we need to carry out behavioral experiments because of the replication crisis in psychology makes it unsafe to assume that we know the qualitative results of humans on our experiments.  After an exchange, the reviewer now agrees that the older classical findings are robust but is still concerned with the robustness of some of the newer studies.  What results does the reviewer question?  All studies in the toolkit have not only been replicated, but we have selected phenomena with large effect sizes. For example, consider the relational vs coordinate change dataset images taken from Hummel & Stankiewicz (1996).  The authors reported 5 replications, and the effects were massive (e.g., in Experiment 1, participants distinguished the relational images 75% and the coordinate images 25%, with similar sized effects across all experiments).  We maintain that all the studies we have included are highly robust.
>
> The reviewer also does not address a logical problem with his criticism.  If indeed there is doubt that the findings we have included in our toolkit replicate, why would he or she want us to collect a single behavioral dataset that researchers are attempt to replicate. Why would our single experimental dataset on each phenomenon be more likely to be a true reflection of human performance compared to the many replicated studies in the literature?  We think the task of qualitatively capturing core findings from psychology that are highly robust is an important approach that needs more attention in the literature.
>
> The novelty of our approach:
>
> Our goal is to challenge researchers in the ML community to adopt a different way of assessing ANN-brain alignment.  The vast majority of research focuses on observational studies, and key findings from the psychological literature are largely ignored.  The reviewers want us to carry out the standard approach of testing a range of models on benchmarks of datasets that assess how much variance is explained. But we want to change this approach, and instead, have researchers systematically test models, more in line with the experimental approaches included in the 9 references we provided to Reviewer wknk.
>
> Reviewer VwBg who is quite positive, now questions whether ICLR is the right venue because it is not clear whether the toolkit address ICLR’s focus on the “novelty of the insights”.  We have specifically submitted to ICLR because our approach is not being widely adopted in the ML community.  In our view, the experiments we are introducing to the ML community and the methods we are advocating are needed to get better insights into ANN-brain alignment. It is the novelty of our approach in this context that is leading to such pushback.
>
> Bowers, J. S., Malhotra, G., Dujmović, M., Montero, M. L., Tsvetkov, C., Biscione, V., ... & Blything, R. (2023a). Deep problems with neural network models of human vision. Behavioral and Brain Sciences, 46, e385.
> Bowers, J. S., Malhotra, G., Adolfi, F., Dujmović, M., Montero, M. L., Biscione, V., ... & Heaton, R. F. (2023b). On the importance of severely testing deep learning models of cognition. Cognitive Systems Research, 82, 101158.
> Hummel, J.E., & Stankiewicz, B. J.,(1996). Categorical relations in shape perception. Spatial vision, 10(3), 201-236.

---

> ### Comment · Reviewer_wknk · 2024-11-28
> **Response**
>
> I appreciate all of these thoughts. Progress in NeuroAI *feels* like it has stalled. I also suspect this is due to an over reliance on models and methods that are designed without any context from psychology and Neuro. Focusing on incremental improvements on BrainScore or whatever benchmark likely won't get us anywhere.
>
> So what's the point of asking for DNN performance on these benchmarks? (1) It could reveal mechanisms that are more/less effective in reproducing different visual psych phenomena. For example, local recurrence (convRNN) vs. FF (convnet) vs. all-to-all attention (ViT). Perhaps there's some insights to be had? (2) The authors have repeatedly claimed that DNNs will struggle to act human-like on these benchmarks. These claims simply must be backed up by experiments. Also, as stated by egun, there should either release human data or if an observational benchmark won't work, they should run and report a perturbation experiment on humans/DNNs that backs up their claim that perturbations are indeed critical.
>
> I (and it seems the other reviewers) are very interested in this line of work. But the authors need to show novelty (not simply cite past work) and empirically back up their claims. Otherwise they are offloading this process to other reviewers to validate if their philosophy and experiments are indeed important for building better models of human vision. Note that the authors are not the first to suggest that models should be able to explain psychological data, either. Many othehrs (including those cited) have compared models with humans on DNNs on these tasks.

---

> > ### Author Response · Authors · 2024-11-28
> >
> > Ours is not a contribution in the form of numerical data, nor is it a philosophical contribution. We certainly didn't invent the experimental manipulation of variables, or decide that such an approach is valuable. Our contribution is to 1) identify that ML is not taking a standard scientific approach 2) review the literature of over a century of robust vision research in psychology to find the minimum set of important phenomena to account for and 3) to provide a flexible set of tools to enable the ML community to use scientific methods to study ANN-human alignment around these phenomena in vision. We believe this is a novel and valuable contribution.

---

### Meta-Review · Area_Chair_oGoF · 2024-12-13

**Metareview:**

The paper introduces the MindSet: Vision toolbox which allows us to manipulate images systematically and uses scripts to test DNNs on 30 psychological experiments. This enables us to carry out hypothesis-driven comparisons with human visual perception.

While the paper is well-written and the motivation to use augmentations for analyzing different behaviors is compelling, it received mixed reviews, with two scores of 8 and three scores of 3. Discussions among the AC and reviewers highlighted key issues that persist despite the rebuttal:

[limited insights from these experiments] The paper lacks systematic benchmarking of all baseline AI models in these experiments. It remains unclear what we should do or not do to improve the designs of these models even after benchmarking models on these experiments.

[the dataset is incomplete] Human behavioral data in these experiments is missing. This makes us fail to assess how much variance models account. Besides, these human performances can serve as an upper bound and help us identify performance gaps between humans and AI models. Unfortunately, all these are missing in the current version.

[limited contributions and novelty claims are vague] Given the two points above, the paper neither contributes a complete solid dataset nor does the paper present claims or insights on what the current AI models are bad or good at.

At this stage, the paper is not ready for publication, but addressing these issues could significantly enhance future versions.

**Additional Comments On Reviewer Discussion:**

The paper has mixed reviews with two 8s, and three 3s.

After reading all the responses, AC agreed with the three reviewers who voted for paper rejection due to the following reasons.

[limited insights from these experiments] The paper lacks systematic benchmarking of all baseline AI models in these experiments. It remains unclear what we should do or not do to improve the designs of these models even after benchmarking models on these experiments.

[the dataset is incomplete] Human behavioral data in these experiments is missing. This makes us fail to assess how much variance models account. Besides, these human performances can serve as an upper bound and help us identify performance gaps between humans and AI models. Unfortunately, all these are missing in the current version.

[limited contributions and novelty claims are vague] Given the two points above, the paper neither contributes a complete solid dataset nor does the paper present claims or insights on what the current AI models are bad or good at.

In the follow-up, the authors provided general arguments to tackle these points; however, they are insufficient and unconvincing without empirical evidence. The authors are encouraged to revise the paper and resubmit to future venues such as neurips dataset and benchmark track.

---

### Decision · Program_Chairs · 2025-01-22

Reject